# Epigenetic suppression of PGC1α (*PPARGC1A*) causes collateral sensitivity to HMGCR-inhibitors within BRAF-treatment resistant melanomas

Jiaxin Liang[1,2], Deyang Yu[1,2], Chi Luo[1,2,4], Christopher Bennett[1,2,5], Mark Jedrychowski[1,2], Steve P. Gygi [2], Hans R. Widlund [3] ✉ & Pere Puigserver [1,2] ✉

While targeted treatment against BRAF(V600E) improve survival for melanoma patients, many will see their cancer recur. Here we provide data indicating that epigenetic suppression of PGC1α defines an aggressive subset of chronic BRAF-inhibitor treated melanomas. A metabolism-centered pharmacological screen further identifies statins (HMGCR inhibitors) as a collateral vulnerability within PGC1α-suppressed BRAF-inhibitor resistant melanomas. Lower PGC1α levels mechanistically causes reduced RAB6B and RAB27A expression, whereby their combined re-expression reverses statin vulnerability. BRAF-inhibitor resistant cells with reduced PGC1α have increased integrin-FAK signaling and improved extracellular matrix detached survival cues that helps explain their increased metastatic ability. Statin treatment blocks cell growth by lowering RAB6B and RAB27A prenylation that reduces their membrane association and affects integrin localization and downstream signaling required for growth. These results suggest that chronic adaptation to BRAF-targeted treatments drive novel collateral metabolic vulnerabilities, and that HMGCR inhibitors may offer a strategy to treat melanomas recurring with suppressed PGC1α expression.

Combinatorial treatment with BRAF and MEK inhibitors constitutes an effective therapy against melanomas harboring BRAF V600-missense mutations[1]. Although most patients respond to this treatment modality, many will see their tumors recur due to cancer cells' ability to evolve and adapt which effectively curbs the long-term survival outlook[2]. This is broadly referred to as therapeutic resistance, or if relatively rare in the initial cancer and when recurrence is seen with a significant delay termed cancer cell persistence. Multiple mechanisms have been identified that contribute to targeted BRAF treatment resistance including: (i) genetic mutations in growth promoting components such as NRAS or MEK that reactivate the MAPK pathway[3,4], as well as hyperactivation of the PI3K/AKT pathway[5,6]; (ii) alternative splicing of the BRAF gene that promotes RAF-dimerization and downstream signaling in the presence of drug[7]; (iii) tumor microenvironment changes such hypoxia or increased secretion of growth factors could desensitize melanoma cells from BRAF inhibitors[8–11]. In addition, melanoma phenotypic switching associated with therapeutic resistance likely involves epigenetic as well as metabolic changes[9,12]. It is currently not entirely known whether targeted BRAF-drug resistance is widespread, or only found pre-existing within

[1]Department of Cancer Biology, Dana-Farber Cancer Institute, Boston, MA, USA. [2]Department of Cell Biology, Harvard Medical School, Boston, MA, USA. [3]Department of Dermatology, Brigham and Women's Hospital, Harvard Medical School, Boston, MA 02115, USA. [4]Present address: Parthenon Therapeutics, Boston, MA 02135, USA. [5]Present address: Atavistik Bio, Cambridge, MA 02139, USA. ✉e-mail: hwidlund@bwh.harvard.edu; pere_puigserver@dfci.harvard.edu

a small population of cancer cells. Importantly, it is not understood whether there are any underlying mechanisms that can be used to inform treatment resistance, and whether co-therapeutic strategies can prevent or target emerging resistance.

We previously identified a melanoma subset defined by the reliance on PGC1α (encoded by *PPARGC1A*), which acts to promote mitochondrial functions and oxidative stress survival[13], including effects from BRAF-targeted treatments[14]. However, high-levels of PGC1α also curb invasive melanoma traits[13,15], thus this transcriptional coactivator and master regulator of mitochondrial biogenesis balances growth and survival with that of invasion and metastatic spread.

## Results

With this scientific background in mind, we sought to explore whether changes in PGC1α expression is related to clinical outcome following BRAF-targeted treatments. Within a combined dataset with clinical outcome containing pre-, on-treatment transcription data (normalized merge of GSE50509, GSE61992, GSE99898), we could discern that within patient biopsies where the baseline *PPARGC1A* expression was above median (high) and reduced by at least two-fold on-treatment, these patients had significantly worse overall survival (Mantel-Cox log rank, $p = 0.05$) compared to the other patients in the combined dataset (Fig. 1a, b).

To model this clinical paradigm in vitro, we assessed five BRAF(V600E)-mutant melanoma cell lines with high PGC1α expression at baseline and subjected them to chronic treatment with the BRAF inhibitor PLX4032[16]. As indicated by substantial decrease in cell growth, PLX4032 BRAF-inhibitor treatment was initially effective, but after 3–5 weeks, proliferating cells emerged in the presence of the drug (Fig. 1c, d; Supplementary Fig. 1a). Although a temporal initial increase in *PPARGC1A* was evident[14], the emerging resistant cells exhibited significantly lower expression of *PPARGC1A*, consistent with the clinical correlative analyses (Fig. 1a). Furthermore, chronic selection with PLX4032 yielded resistance also to a different BRAF inhibitor, dabrafenib (Supplementary Fig. 1b). Chronic PLX4032 treated G361 and SKMEL5, however, yielded resistant cells with elevated *PPARGC1A* expression (Supplementary Fig. 1c), which parallels the other end of the change-in-expression continuum seen across melanoma biopsies pre-/post-BRAF inhibitor treatment (Fig. 1a). In contrast to parental cells, chronic-treated BRAF-inhibitor resistant melanoma cell lines did not decrease phospho-ERK1/2 or phospho-S6 levels when subjected to BRAF inhibitor (Supplementary Fig. 1a). Akin to genetic suppression of PGC1α[15], chronic BRAF-inhibitor treated A375P, K029A, and SKMEL3 cells increased migratory and invasive measures using in vitro and in vivo assays (Fig. 1e; Supplementary Fig. 1d, e), and in a manner that was reversed by ectopic PGC1α expression (Fig. 1f; Supplementary Fig. 1f). Collectively these data suggest that the aggressive melanoma phenotypic traits were dependent on *PPARGC1A* suppression as a result of chronic BRAF inhibitor treatment, which is consistent with our prior studies[15].

To determine the mechanism by which PGC1α levels were reduced following chronic BRAF inhibitor treatment, we assessed changes in epigenetic histone marks across the *PPARGC1A* promoter region and found an increase in repressive H3K27me3 and a decrease in active H3K27ac marks (Fig. 1g; Supplementary Fig. 2a). Interestingly, H3K27me3 marks are catalyzed by EZH2 and we found that EZH2 expression was upregulated in chronic BRAF-inhibitor treated cells wherein *PPARGC1A* levels decreased (Supplementary Fig. 2b), thus emphasizing the role of EZH2 on *PPARGC1A* silencing[16]. Treatment with the EZH2-inhibitor tool compound 3-deazaneplanocin A (DZNep)[17] or the clinically approved tazemetostat[18], increased PGC1α expression (Fig. 1h; Supplementary Fig. 2c, d), and functionally decreased melanoma cell migration (Supplementary Fig. 2e). Collectively, these results suggest that in a subset of melanoma cells,

chronic BRAF inhibitor treatment can cause epigenetic adaptation that involves *PPARGC1A* suppression and acquisition of aggressive melanoma traits.

Because there is an outstanding clinical need to identify novel and exploitable vulnerabilities that arise within treated BRAF-inhibitor resistant melanomas, we performed a cell-based growth inhibitory chemical screen. Since *PPARGC1A* silencing is expected to alter metabolic demands through impeding mitochondrial biogenesis, we used a metabolism-focused compound library to seek differential sensitivity between parental (drug naïve) and chronic treated BRAF-inhibitor resistant K029A melanoma cells (Fig. 2a). The top hits in this screen using 72 h treatment followed by CCK-8 readout were HMG-CoA reductase (HMGCR) inhibitors, *i.e.* pitavastatin, rosuvastatin, and atorvastatin; known collectively as statins (Fig. 2b), which are commonly used to treat hypercholesterolemia. Indeed, statins have been shown to inhibit the growth of melanoma and many other cancer types in vitro as well as in vivo[19–21]. We validated the increased sensitivity to pitavastatin in resistant K029A, GI50 (resistant) 0.70 μM, (range 0.66-0.75) compared to parental, (GI50 (parental) 8.0 μM), (range 6.8-10.3 μM) (Fig. 2c), which effects were also recapitulated using atorvastatin and lovastatin (Supplementary Fig. 3a, b). Similar increase in HMGCR inhibitor sensitivity was also in chronic BRAF-inhibitor treated A375P and SKMEL3 cells (Supplementary Fig. 3c, d). In addition, chronic treatment of A375P using a combination of BRAF- and MEK-inhibitors similarly yielded cells with suppressed *PPARGC1A* expression and increased statin sensitivity (Supplementary Fig. 3e, f). In contrast, chronic treated BRAF-inhibitor (or combination BRAF- and MEK-inhibitor) resistant G361 and SKMEL5 cells, exhibiting increased PGC1α levels, did not display heightened statin sensitivity (Supplementary Fig. 3e, g–i). Furthermore, genetic manipulation of *PPARGC1A* using Cas9/sgRNA editing in parental (treatment naïve) cells or ectopic expression of PGC1α in chronic adapted BRAF-inhibitor resistant K029A cells and A375P cells, increased and decreased sensitivity to statin treatment, respectively (Fig. 2d, e; Supplementary Fig. 3j–m). Similarly, EZH2-inhibitor treatment reduced the sensitivity to statins within chronic adapted BRAF-inhibitor resistant cells (Fig. 2f).

To determine whether statin treatment in vivo could evidence a therapeutic opportunity to target BRAF-inhibitor resistant melanoma cells, we subjected established tumors from chronic treated BRAF-inhibitor resistant K029A cells (K029A-R), A375P cells (A375P-R) and SKMEL3 cells (SKMEL3-R) to pitavastatin. The rationale behind treating xenografted tumor models with pitavastatin alone without BRAF inhibitor was that the sensitivity to statin was not BRAF inhibitor dependent (Supplementary Fig. 3n). Pitavastatin treatment showed significant growth inhibition compared to mock treatment (Fig. 2g–i). Similarly, established tumors from chronic BRAF + MEK inhibitor treatment resistant A375P (A375P-DR) cells also displayed potent growth retardation (Fig. 2j). Consistently, no effect from statin treatment was seen on parental (treatment naïve) tumors (Fig. 2k; Supplementary Fig. 3 o–q). These data clearly suggest that reduced PGC1α expression, driven by chronic adaptation to BRAF(+ MEK) inhibitor treatment, provokes sensitivity to HMGCR inhibitors.

Because HMGCR is the rate limiting enzyme in the mevalonate pathway that fuels biosynthesis of cholesterol and isoprenoids (Supplementary Fig. 4a), we investigated if certain metabolite intermediates could be responsible for the observed statin sensitivity. As expected, mevalonate rescued cell growth inhibition caused by statin treatment, indicating that observed effects were on target (Fig. 3a). Among the downstream metabolites, only the isoprenoid geranylgeranyl pyrophosphate (GGPP) rescued statin-induced cell growth inhibition, while cholesterol, CoQ and farnesol did not (Fig. 3b; Supplementary Fig. 4b–d), which is consistent with prior reports suggesting that GGPP can bypass statin-mediated cancer cell growth inhibition[22,23].

Since GGPP is used to prenylate proteins, among which the RAB family is predominant[24,25], we performed mass spectrometry-based proteomic analysis[26] of chronic BRAF-inhibitor treated cells in an attempt to identify changes pertaining to specific RAB proteins. Interestingly, we found that chronic BRAF-inhibitor treated K029A cells had downregulated the levels of *RAB6B* and *RAB27A* compared to their parental counterparts (Fig. 3c, Supplementary Fig. 4e). We accordingly surveyed publicly available expression level data for melanoma biopsies (TCGA: SKCM; Fig. 3d) and cell lines (Broad: CCLE

21Q1: Supplementary Fig. 4f) and found that *RAB6B* and *RAB27A* expression highly correlated with *PPARGC1A* levels. In agreement, chronic treated BRAF-inhibitor resistant A375P, K029A, and SKMEL3 melanoma cells exhibited reduced expression of these two RAB genes (Fig. 3e; Supplementary Fig. 5a).

To determine whether HMGCR-inhibitor sensitivity in the chronic treated BRAF-inhibitor resistant cells was directly linked to reduced levels of RAB6B and RAB27A, we used lentiviral transduction to express each of RAB6B and RAB27A or a combination of both.

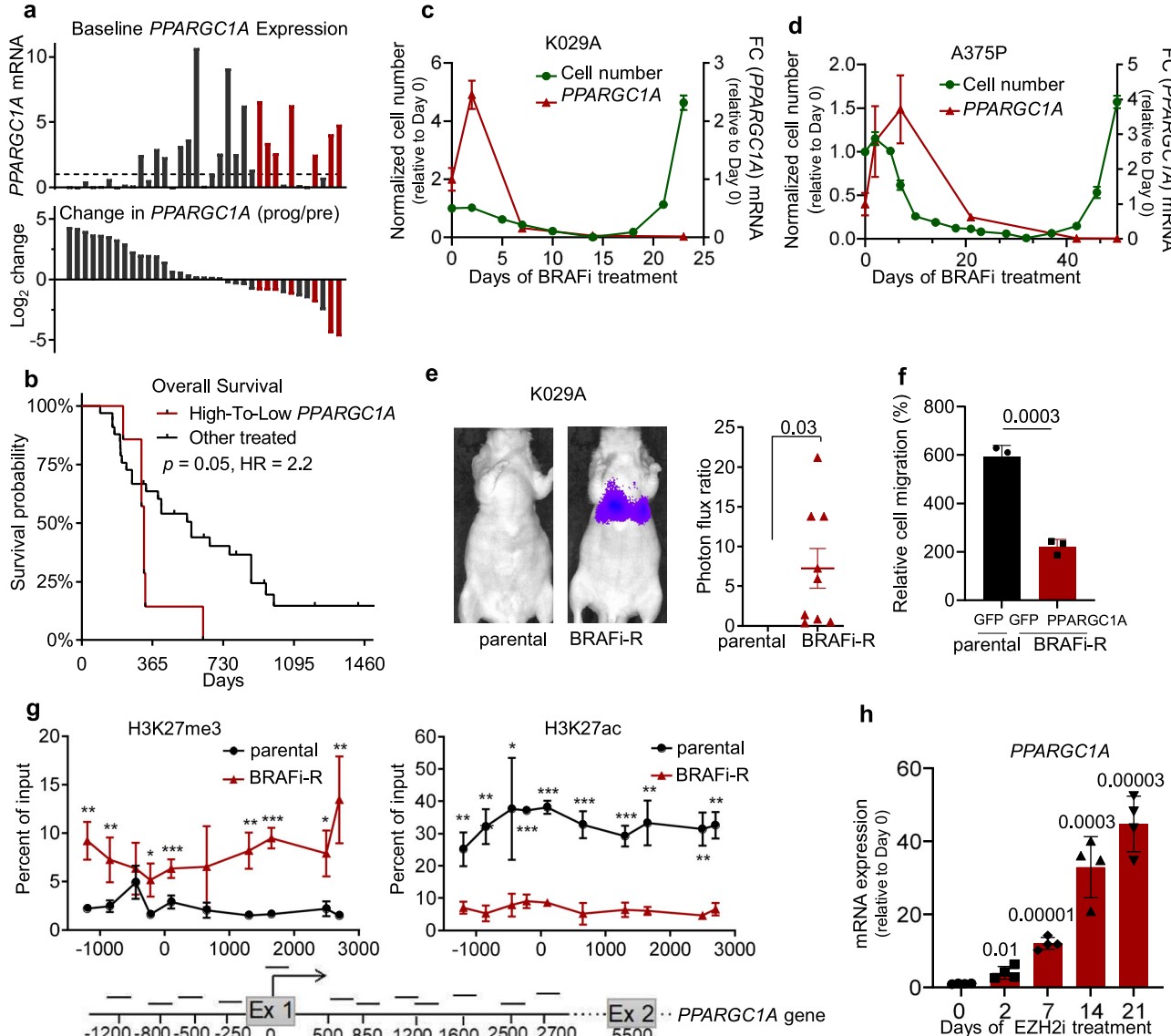

**Fig. 1 | A subset of BRAF-inhibitor resistant melanomas silence PPARGC1A expression. a** Changes in PPARGC1A expression across melanoma patient biopsies at baseline (pre-treatment) and changes baseline (pre-treatment) to time at progression using quantile normalized merge of the publicity available datasets GSE50509, GSE61992, GSE99898 curated for redundant patient occurrence. **b** Survival analysis (Mantel-Cox log rank; *p* = 0.05) from BRAF-inhibitor treatment as stratified based on changes in PPARGC1A expression within the merged (GSE50509, GSE61992, GSE99898) dataset. **c** Fold-change in cell numbers (green) and PPARGC1A mRNA expression levels (red) upon chronic treatment of K029A cells with BRAF inhibitor (1 μM PLX4032) (mean ± SEM, *n* = 3). **d** Fold-change in cell numbers (green) and PPARGC1A mRNA expression level (red) upon chronic treatment of A375P cells with BRAF inhibitor (1 μM PLX4032) (mean ± SEM, *n* = 3). **e** Representative in vivo chemiluminescent imaging of parental or chronic treated K029A BRAF-inhibitor resistant cells (left) with quantification of photon flux ratio

(right) across samples. Quantified values represent mean ± SEM (*n* = 6 for parental; *n* = 9 for BRAFi-R). Significance calculated as un-paired, two-sided *t* test. **f** Quantification of trans-well migration assay data from parental and chronic treated BRAF-inhibitor adapted K029A cells, and reversal using ectopic PGC1α (mean ± SEM, *n* = 3). Significance calculated as un-paired, two-sided *t* test. **g** ChIP analysis of H3K27me3 and H3K27ac marks across the PPARGC1A promoter region in parental and chronic treated BRAF-inhibitor resistant K029A cells (*p* value from left to right for H3K27me3 are 0.004, 0.025, 0.47, 0.023, 0.007, 0.14, 0.003, 0.0002, 0.0166, 0.01 while *p* value for H3K27ac are 0.004, 0.001, 0.03, 0.00003, 0.00002, 0.0009, 0.0005, 0.0026, 0.0009, 0.0006) (mean ± SEM, *n* = 3) (\**p* < 0.05, \*\**p* < 0.001, \*\*\**P* < 0.0001). Significance calculated as un-paired, two-sided *t* test. **h** Normalized PPARGC1A expression across a EZH2 inhibitor (1 μM, DZNep) treatment time course within chronic BRAF-inhibitor adapted K029A cells (two-sided un-paired *t* test, mean ± SEM, *n* = 3).

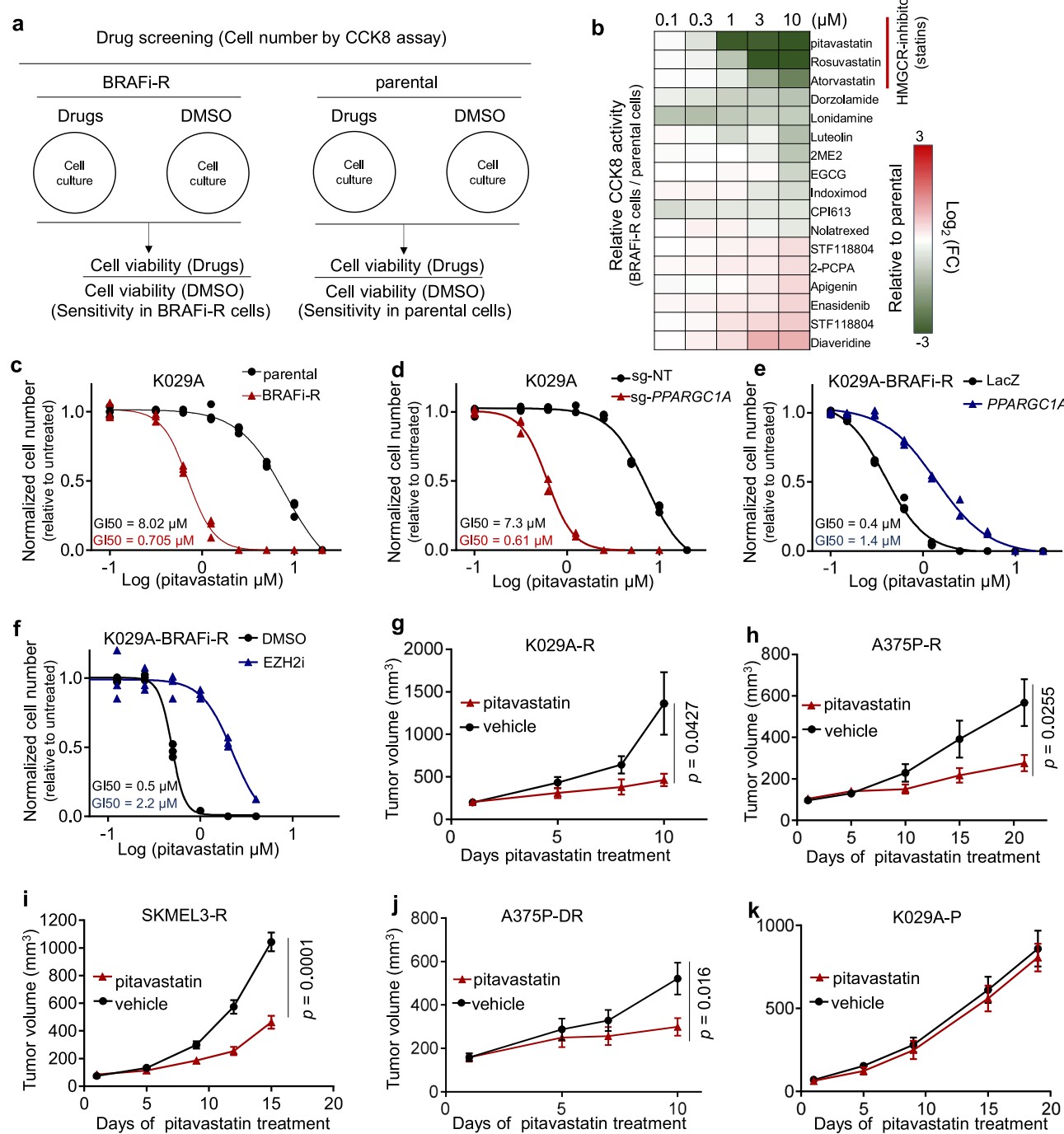

**Fig. 2 | Chronic BRAF-inhibitor treated melanoma cells with silenced PPARGC1A are sensitive to HMGCR inhibitors. a** Schematic depiction of the metabolism-focused small compound screening to identify molecules that selectively block the growth of chronic BRAF-inhibitor treated cells. **b** Log₂ (fold change) in cell viability (chronic BRAF-inhibitor treated vs parental cells), at the indicated concentrations of the listed compounds for 72 h. **c** GI50 of pitavastatin in parental and chronic BRAF-inhibitor treated K029A cells (72 h treatment) (*n* = 3). **d** GI50 of pitavastatin in K029A cells with/without PGC1α depletion (72 h treatment) (*n* = 3). **e** GI50 of pitavastatin in chronic BRAF-inhibitor treated K029A cells with/without ectopic PGC1α expression (72 h treatment) (*n* = 3). **f** GI50 of pitavastatin in chronic BRAF-inhibitor treated K029A cells with or without adaptation to EZH2 inhibitor (1 µM DZNep for 3-week) (*n* = 3). **g** Growth of tumors established from chronic BRAF-inhibitor treated K029A cells (s.c. xenografts in nu/nu mice). Pitavastatin was administrated at 1 mg/kg b.i.d., (mean ± SEM, *n* = 5). Significance calculated as

un-paired, two-sided *t* test. **h** Growth of tumors established from chronic BRAF-inhibitor treated A375P cells (s.c. xenografts in nu/nu mice). Pitavastatin was administrated at 1 mg/kg b.i.d., (mean ± SEM, *n* = 10). Significance calculated as un-paired, two-sided *t* test. **i** Growth of tumors established from chronic BRAF-inhibitor treated SKMEL3 cells (s.c. xenografts in nu/nu mice). Pitavastatin was administrated at 1 mg/kg b.i.d., (mean ± SEM, *n* = 5). Significance calculated as un-paired, two-sided *t* test. **j** Growth of tumors established from chronic BRAF-inhibitor and MEK-inhibitor treated A375P cells (s.c. xenografts in nu/nu mice). Pitavastatin was administrated at 1 mg/kg b.i.d., (mean ± SEM, *n* = 10). Significance calculated as un-paired, two-sided *t* test. **k** Growth of tumors established from parental K029A cells (s.c. xenografts in nu/nu mice)/ Pitavastatin was administrated at 1 mg/kg b.i.d., (mean ± SEM, *n* = 5). Significance calculated as un-paired, two-sided *t* test.

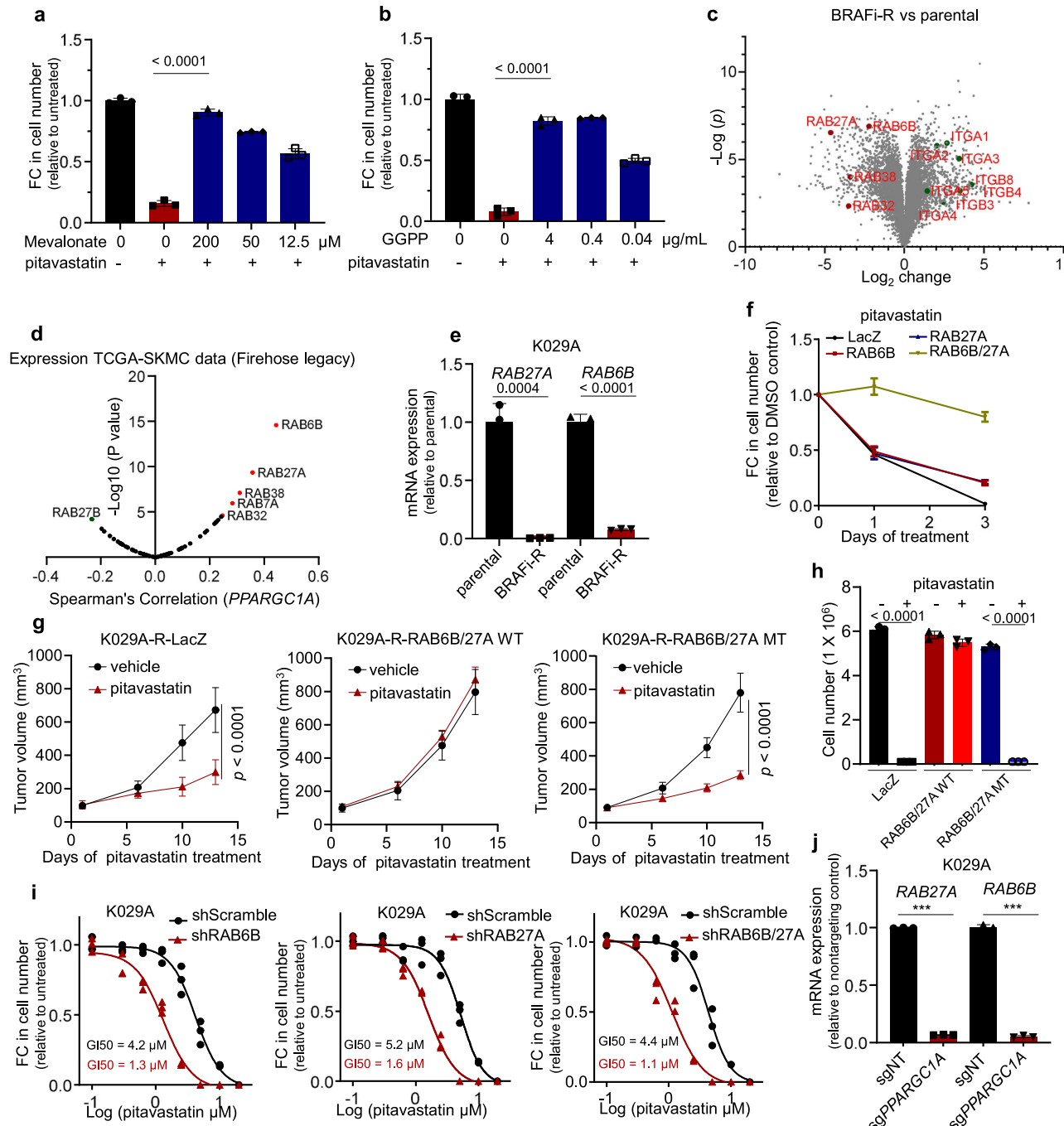

**Fig. 3 | Levels of RAB6B and RAB27 affect sensitivity to HMGCR-inhibitors.**
**a** Change in cell numbers of chronic BRAF-inhibitor treated K029A cells treated with 1 μM pitavastatin in combination with mevalonate at indicated concentration for 72 h (*n* = 3). Significance calculated as un-paired, two-sided *t* test. **b** Change in cell numbers of chronic BRAF-inhibitor treated K029A cells treated with 1 μM pitavastatin in combination with GGPP at indicated concentration for 72 h (*n* = 3). Significance calculated as un-paired, two-sided *t* test. **c** Volcano plots of mass spectrometry assessed protein amounts within parental and chronic BRAF-inhibitor treated K029A cells. Fold-change in proteins (chronic treated vs parental) and *p* value (calculated as unpaired two-sided *t* test) are shown. **d** Gene expression correlation between RAB family members and PPARGC1A within melanoma patient samples (calculated as unpaired two-sided spearman's correlation analysis) (TCGA, Firehose Legacy). **e** Expression of RAB27A and RAB6B in parental and chronic BRAF-inhibitor treated K029A cells (mean ± SEM, *n* = 3). Significance calculated as un-paired, two-sided *t* test. **f** Change in cell numbers of chronic BRAF-inhibitor treated K029A cells with ectopic expression of LacZ, RAB27A, RAB6B or combinatorial

RAB6B and RAB27A in the presence of 1 μM pitavastatin. Fold changes were calculated by the comparison of the cell numbers in pitavastatin treated to the cell numbers in DMSO control (mean ± SEM, *n* = 3). **g** Effect of pitavastatin treatment on tumor s.c. xenograft growth in nu/nu mice comparing chronic BRAF-inhibitor treated K029A cells with overexpression of LacZ, combinatorial wild-type RAB6B and RAB27A, combinatorial prenylation mutant RAB27A and RAB6B, respectively. Mice were treated with pitavastatin 1 mg/kg b.i.d. (mean ± SEM, *n* = 5). Significance calculated with un-paired two-sided t test at the last time point. **h** Cell numbers of parental K029A cells with ectopic expression of LacZ (control), prenylation mutant RAB6B and RAB27A, and wild type RAB6B and RAB27A respectively. Cells were treated with 1 μM pitavastatin for 72 h (mean ± SEM, *n* = 3). Significance calculated as un-paired, two-sided *t* test. **i** GI50 of pitavastatin in parental K029A cells with knockdown of RAB6B, RAB27A or combinatorial knockdown of RAB6B and RAB27A (72 h treatment) (*n* = 3). **j** Expression of RAB27A and RAB6B in parental K029A cells with/without CRISPR/Cas9 sgPPARGC1A targeted editing (mean ± SEM, *n* = 3). Significance calculated as un-paired, two-sided *t* test.

While ectopic expression of RAB6B, or RAB27A, partially reversed the observed statin sensitivity, co-expression of RAB6B and RAB27A essentially rescued growth inhibition by statin treatment, as demonstrated using in vitro cell proliferation and in vivo tumor xenograft growth measures (Fig. 3f, g; Supplementary Fig. 5b, c). In addition, ectopic expression of both RAB6B and RAB27A also rescued statin-induced growth inhibition in K029A-sg*PPARGC1A* cells (Supplementary Fig. 5d–e). We also generated RAB6B and RAB27A alleles defective in the C-terminal prenylation-acceptor sites that were found unable to reverse statin sensitivity (Fig. 3g, h). Further, these effects were specific to combinatorial RAB6B and RAB27A over-expression, because individual RAB7A, RAB32, RAB38 expression, or combination with RAB6B, did not affect statin sensitivity (Supplementary Fig. 6a, b). Conversely, shRNA was used to suppress RAB6B and RAB27A expression in parental (treatment naïve) cells and revealed that each individually provoked a HMGCR-inhibitor sensitivity, that was increased with depletion of both RABs; reaffirming that RAB6B and RAB27A are required for growth in the presence of HMGCR-inhibitors (Fig. 3i; Supplementary Fig. 6c). Overexpression of wild type RAB6B + RAB27A, but not prenylation mutants, also rescued chronic BRAF-inhibitor treated A375P cells from statin sensitivity (Supplementary Fig. 6d, e). These results, taken together, indicate that appropriate RAB6B and RAB27A levels and their subsequent prenylation define the growth suppressive effect of HMGCR-inhibitor treatment in this subset of BRAF-inhibitor resistant melanoma cells.

We next asked if PGC1α participated in the regulation of RAB6B and RAB27A expression. Targeted disruption of *PPARGC1A* using CRISPR/Cas9 resulted in lower RAB6B and RAB27A expression (Fig. 3j; Supplementary Fig. 7a), and conversely, ectopic expression of PGC1α in chronic BRAF-inhibitor treated K029A increased expression of RAB6B and RAB27A (Supplementary Fig. 7b). Emphasizing the tight relationship between the melanocyte master-regulator MITF and PGC1α, the co-activator for MITF activation[27], RAB6B and RAB27A levels decreased following CRISPR/Cas9 targeted *MITF*-deletion (Supplementary Fig. 7c), and ectopic MITF expression restored RAB6B and RAB27A in MITF-deleted cells (Supplementary Fig. 7d). Towards the observed HMGCR-inhibitor sensitivity, targeted deletion of either PGC1α or MITF sensitized the cells to pitavastatin but could not compensate for one another (Supplementary Fig. 7e, f), suggesting that each PGC1α and MITF participated in controlling HMGCR-inhibitor sensitivity, likely by co-regulating RAB6B and RAB27A levels.

We next wanted to determine what inherent processes within chronic treated BRAF-inhibitor resistant melanoma cells were affected by HMGCR-inhibitor treatment. Mass spectrometry-based proteomic analyses of cells treated with pitavastatin, compared to mock-treatment, identified changes associated with mitotic cell division and DNA damage (Fig. 4a, b; Supplementary Fig. 8a). Notably, these differences were dependent on prenylation of RAB proteins because addition of GGPP or ectopic RAB6B + RAB27A expression obviated these changes (Fig. 4b). In agreement, fluorescence activated cell sorting (FACS) analysis indicated a robust accumulation of cells in G0/G1 following statin treatment of chronic adapted cells (Fig. 4c). Likewise, a statin-induced DNA damage response was confirmed using γ-H2AX detection (Supplementary Fig. 8b). Furthermore, while statin treatment induced PARP1 cleavage in chronic treated BRAF-inhibitor resistant cells, only co-expression of RAB6B and RAB27A could abolish these markers of cell death (Supplementary Fig. 8c).

To explore how RAB6B and RAB27A levels affect HMGCR inhibitor sensitivity, we analyzed differential mass spectrometric-based proteome profiles of parental and chronic BRAF-inhibitor treated melanoma cells. These analyses revealed an increase in membrane and extracellular matrix-associated proteins (Supplementary Fig. 4e, 8a). Firmly aligned with our previous work on how reduced PGC1α levels

promotes increased metastatic behavior of melanoma cells[15], integrin receptors were here also found upregulated in the chronic treated BRAF-inhibitor resistant melanoma cells (Fig. 1e; Supplementary Fig. 4e), and accordingly determined to be PGC1α-dependent at the RNA level (Supplementary Fig. 9a), but unaffected by HMGCR-inhibitor treatment (Supplementary Fig. 9b).

Because RAB proteins are critical regulators of membrane traffic[28], including that of integrin receptors, we used cell fractionation to examine whether HMGCR-inhibitor caused a redistribution of RAB6B and RAB27A, compared to C-terminal prenylation-defective allele variants. In comparison to mock-treatment, we found that pitavastatin caused a redistribution of the wild-type RAB-alleles into the soluble cell fraction where the prenylation-site mutant alleles resided (Fig. 4d; Supplementary Fig. 10a), clearly indicating that HMGCR-inhibition within these cells impacts RAB localization dependent on prenylation. Using mass spectrometry-based proteomic analyses of biotin-purified cell surface proteins, we next assessed the extent to which HMGCR-inhibitor treatment affected, and RAB6B + RAB27A expression rescued, proteins located at the cell surface. This approach revealed that pitavastatin treatment reduced cell surface localization of a large number of integrins within the chronic treated BRAF-inhibitor resistant cells, which were correspondingly retained by RAB6B and RAB27A over-expression (Fig. 4e; Supplementary Fig. 10b). As confirmation of the proteomic analyses, we used biotin-purification of surface proteins by Western blot analysis, compared to total proteins, to verify alterations in cell surface abundance of α5 integrins modulated by sg*PPARGC1A* (yielding an increase) and HMGCR-inhibitor treatment (resulting in a decrease) (Fig. 4f, and α4, αV and β4 integrins: Supplementary Fig. 10c). As expected, over-expression of RAB6B and RAB27A reversed the decrease in cell surface localization of integrins following HMGCR-inhibitor treatment (Fig. 4f). Furthermore, HMGCR-inhibitor treatment also affected downstream phospho-FAK signaling in cells with reduced PGC1α using sg*PPARGC1A*, or chronic treated with BRAF-inhibitor, which could be rescued by ectopic RAB6B + RAB27A or GGPP supplementation (Fig. 4g; Supplementary Fig. 10d). The effect of HMGCR-inhibitor on phospho-FAK signaling was also observed in the tumors derived from K029A-R cells (Supplementary Fig. 10e). In contrast, the phospho-FAK levels were not changed in chronic treated BRAF-inhibitor resistant G361 and SKMEL5 cells upon HMGCR-inhibitor treatment (Supplementary Fig. 10f).

Since integrins are activated at sites of extracellular matrix binding[29], and downstream activated FAK-signaling is able to promote survival of cells even in the absence of attachment[30], we assessed these two functional endpoints. Chronic treated BRAF-inhibitor resistant cells pre-treated with HMGCR inhibitor were unable to attach to cell culture plates, whereas parental or ectopic of RAB6B + RAB27A in the BRAF-inhibitor resistant enabled attachment independent of HMGCR-inhibitor pre-treatment (Fig. 4h). Next, we assessed whether the elevated FAK signaling within the BRAF-inhibitor resistant cells impacted resistance to detachment-induced apoptosis (anoikis) of cells maintained in suspension, and if this could be modulated by supplementation using the antioxidant N-acetyl cysteine (NAC) as previously described[31] or HMGCR inhibitor (Fig. 4i, j). Chronic treated BRAF-inhibitor resistant cells, independent of ectopic RAB6B + RAB27A, were more resilient against anoikis as compared to their parental counterparts, but the latter improved to the same extent by supplementing the media with NAC, suggesting antioxidants could overcome a metabolic defect[31] caused by loss of matrix attachment not perceived by the BRAF-resistant cells. Interestingly, also SKMEL5 and G361 cells, independent of their chronic treatment with BRAF-inhibitor, improved their resistance to anoikis by NAC supplementation (Supplementary Fig 11a, b). Towards HMGCR inhibitor sensitivity, only chronic treated BRAF-resistant cells PGC1α-silenced cells without combined ectopic RAB6B and RAB27A displayed diminished survival in suspension upon

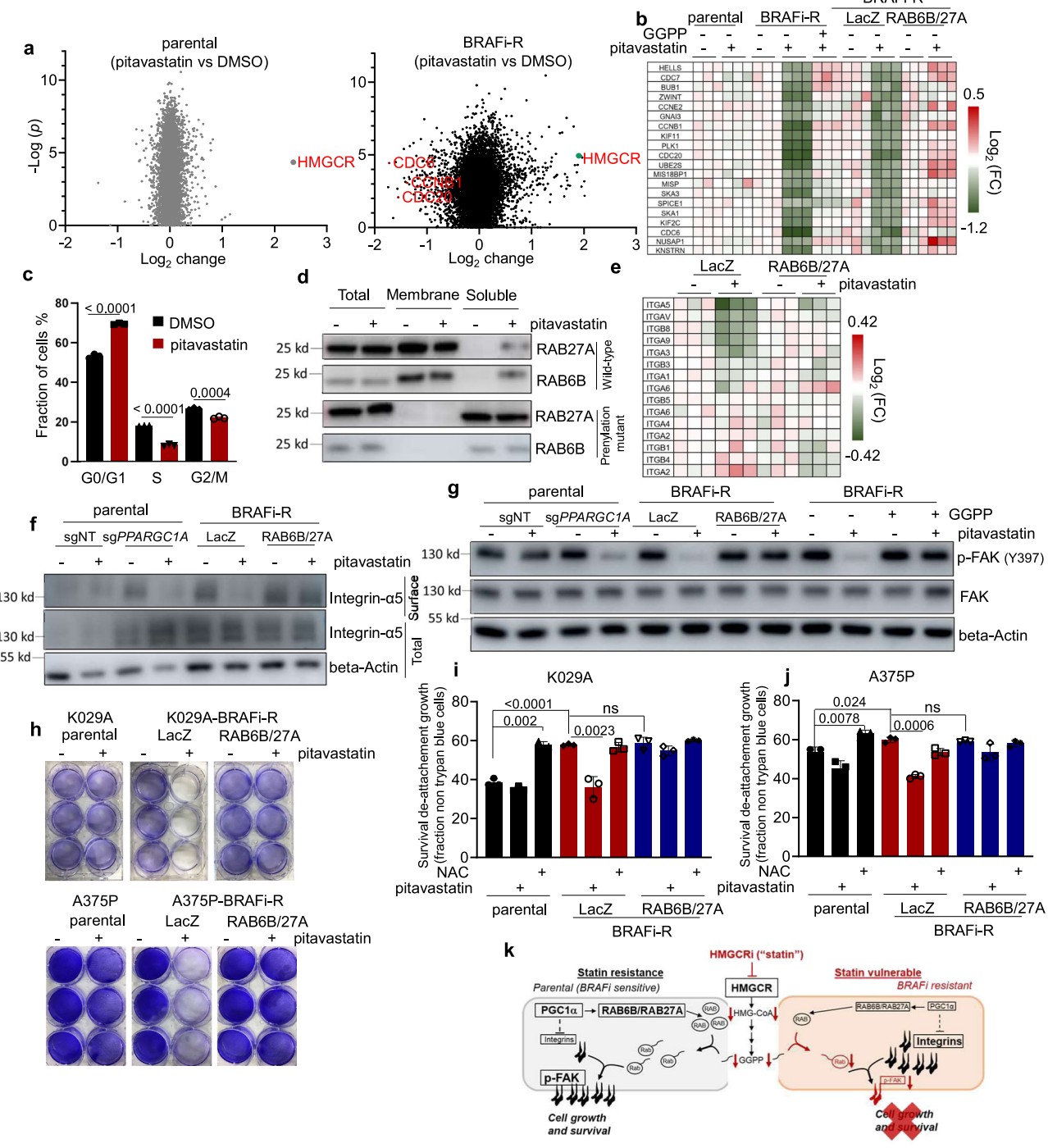

treatment (Fig. 4i, j). We also used matrigel, which is known to activate integrin receptor and is a major constituent of the extracellular matrix, to assess changes in FAK activation of cells maintained in mono-cellular suspension (within methylcellulose). As predicted, matrigel rescued anchorage-independent survival in mock treated cells, but the striking difference was with HMGCR-inhibitor. HMGCR-inhibitor treatment did not affect anoikis within parental cells, or chronic-treated BRAF-inhibitor resistant cells with ectopic RAB6B and RAB27A, but yet resistant cells exhibited increased anoikis in these assays (Supplementary Fig 11c). Furthermore, the observed anoikis sensitivity closely followed integrin-dependent phospho-FAK levels (Supplementary Fig. 11d). These data indicates that resistance to anoikis driven by elevated integrin receptor expression and FAK signaling is functionally separable from the collateral sensitivity to HMGCR inhibitors

seen in chronic treated BRAF inhibitor resistant melanoma cells with silenced PGC1α expression.

## Discussion

Collectively, this work has identified a subset of BRAF inhibitor resistant melanomas defined by epigenetic silencing of PGC1α that emerge during chronic treatment, with metastatic traits (invasion, anchorage independent survival, and in vivo metastatic spread), but acquire a collateral vulnerability towards HMGCR-inhibitor treatment. Suppression of PGC1α results in lower RAB6B and RAB27A levels that in response to HMGCR-inhibitor treatment reduces cell surface trafficking of integrin and downstream FAK signaling, which is functionally displayed as a specific and collateral vulnerability in these chronic treated BRAF-inhibitor resistant cells (Fig. 4k). The growth inhibitory

**Fig. 4 | Pitavastatin induces cell death through the inhibition of prenylation of Rabs and the disruption of integrin signal pathway. a** Volcano plot of proteomic data. Protein changes in parental (left) and chronic BRAF-inhibitor treated K029A cells (right) upon treatment with 1 μM pitavastatin for 24 h. *p* value (two side unpaired *t* test) plotted in a log10 scale, including fold-change in detected protein (pitavastatin vs DMSO) plotted in a log2 scale. Significance calculated as un-paired, two-sided *t* test. **b** Heatmap of the expression of cell cycle associated genes in K029A parental, chronic BRAF-inhibitor treated, chronic BRAF-inhibitor treated with ectopic expression of LacZ or dual Rabs, following treatment with/without 1 μM pitavastatin for 24 h. As indicated, one group of chronic BRAF-inhibitor treated K029A cells were added GGPP at 0.4 μg/ml. **c** Fractions of G0/G1, S and G2/M phase cells in K029A chronic treated BRAF-inhibitor adapted cells. Cells were treated with/without pitavastatin [1 μM for 24 h] (mean ± SEM, *n* = 3). Significance calculated as un-paired, two-sided *t* test. **d** Western blots of RAB6B and RAB27A. Total, membrane attached and soluble RAB6B and RAB27A in K029A chronic treated BRAF-inhibitor adapted cells with the ectopic expression of wild type dual RABs or prenylation-mutant dual RABs are indicated. Cells were treated with/without pitavastatin [1 μM for 24 h]. **e** Heatmap of changes in plasma membrane associated integrins within chronic treated BRAF-inhibitor treated K029A cells with ectopic expression of RAB6B and RAB27A (LacZ as a control). Cells were treated with/without pitavastatin [1 μM for 18 h]. **f** Western blot analyses of plasma membrane located integrin-α5 and integrin-β1 in parental K029A cells with/without PGC1α knockout and chronic BRAF-inhibitor treated K029A cells with ectopic expression of LacZ or combination of RAB6B and RAB27A. The cells were treated with pitavastatin [1 μM for 18 h]. **g** Western blots of FAK-pY397 in K029A parental cells with/without PGC1α knockout and chronic treated BRAF-inhibitor adapted cells with the ectopic expression of LacZ or RAB27A and RAB6B. The cells were treated with pitavastatin [1 μM for 24 h]. **h** Crystal violet staining images. K029A (left) and A375P (right) parental, chronic BRAF-inhibitor adapted cells with ectopic expression of LacZ or combination RAB6B and RAB27A were pre-treated with pitavastatin for 18 h and then seeded into 6-well plates. The unattached cells were washed away with PBS and the attached cells subsequently stained with crystal violet. **i, j** Percentages of viable cells in suspension. K029A (**i**) and A375P (**j**) parental, chronic BRAF-inhibitor adapted cells with ectopic expression of LacZ or RAB6B and RAB27A were suspended in 0.5% methylcellulose and treated with/without NAC or pitavastatin. Viable cells were assessed by staining negative for trypan blue (mean ± SEM, *n* = 3). Significance calculated as un-paired, two-sided *t* test. **k** A proposed schematic model depicts how chronic BRAF-inhibitor treatment causes downregulation of PPARGC1A with effects on RAB6B and RAB27A, which causes a collateral vulnerability towards HMGCR inhibitors (statins) by elevated dependence on RAB-mediated integrin trafficking.

effect of statins in *PPARGC1A* silenced cells can be rescued by mevalonate or GGPP supplementation, or combinatorial expression of wild-type alleles of RAB6B and RAB27A. RAB proteins are key regulators of endosomal trafficking and linked to localization of proteins to plasma membrane[32–35]. RAB proteins require prenylation of the C-terminal CAAX using GGPP – a covalently attached lipid-moiety that allows membrane insertion[36,37]. Based on our data, we surmise that BRAF-inhibitor resistant cells display a collateral vulnerability to HMGCR inhibition because the pool of GGPP is used by other RAB proteins to traffic a higher abundance of integrins to and from the plasma membrane, which promotes anoikis resistance and metastatic traits. In the presence of HMGCR inhibitor (statin), the already reduced amounts of RAB6B and RAB27A becomes functionally limiting due to insufficient prenylation with diminishing amounts of membrane associated integrin complexes and attenuated FAK signaling (Fig. 4k).

Previous studies in melanoma have suggested that there is an inherent barrier to metastatic spread that involves excessive oxidative stress, which can be overcome by antioxidant treatment[38]. Reducing PGC1α-levels would be expected to increase oxidative stress by compromising mitochondrial biogenesis as well as anti-oxidant scavenging and which creates a vulnerability in combination with BRAF-inhibitor treatment[14]. Akin to the to the Warburg effect[39], attenuation of PGC1α-function in melanoma cells, and their dependence on oxidative phosphorylation, causes a shift towards a metastatic phenotype through relieving transcriptional suppression of integrin receptors and reduced catabolism[15]. It is within this subset of melanoma cells defined by high PGC1α levels and elevated oxidative phosphorylation where we find that chronic-adaptation to BRAF-inhibitor treatment leads to emergence of an alternative epigenetic-state that involves silencing of PGC1α-function and upregulation of integrin receptors. This epigenetic-state is at least partly reversible because EZH2-inhibitor treatment relieve silencing of PGC1α expression. Integrin receptor activation and downstream FAK activity promotes survival and metastatic spread[40]. Antioxidant treatment promotes metastatic spread[38] and resistance to anoikis[31], while subjecting melanoma cells with heightened reliance on catabolism to chronic BRAF-inhibitor treatment allows an alternate epigenetic fate that involves attenuated mitochondrial function with a shift towards reliance on extracellular matrix-dependent processes; however with compromised ability to respond to certain metabolic insults including HMGCR inhibition.

Identifying collateral tumor cell vulnerabilities that arise from acquired treatment resistance[41,42] constitute an attractive means for developing combinatorial strategies that may extend the response to existing oncogene-targeted therapies. For treatment of melanoma with targeted BRAF inhibitors, adaptive mechanisms have been suggested to involve metabolic changes, such as increased dependency on lipid metabolism[43,44]. While there are preclinical studies suggesting that HMGCR inhibitors alone, or in combination with other drugs, effectively inhibit melanoma cell growth[45,46], the usefulness of statins for melanoma patients is still controversial[47–49]. Identification of genuine molecular markers that define or predict the sensitivity to statin are still needed. Our studies support the use of changes in PGC1α expression as a potential marker for statin sensitivity. These findings indicate an opportunity to initiate selective studies melanoma patients to assess whether HMGCR inhibitor treatment with would be efficacious at BRAF-inhibitor progression. Furthermore, these observations might also have ramifications for other tumor-types that depend on RAB function, or potentially inform mechanisms involved in causing HMGCR-inhibitor toxicities in non-malignant cells, independent of cholesterol synthesis inhibition[50,51].

## Methods

### Data reporting

For animal experiments, we estimated the response to be more than two-fold reduction compared to untreated, and to overall reach a sought alpha of 0.05 at a 80% power (FDR *q* < 0.2), we assumed that the K029A model would be attached with Standard Deviations of 25% (untreated) and 50% (treated) resulting in a needed cohort size of 4, and for A375P Standard Deviations of 50% (untreated) and 50% (treated) resulting in a cohort size of 10. There was no randomization of cohorts, and the investigators were not blinded to allocation or outcome assessment.

### Analyses of clinical melanoma datasets

The publicly available datasets GSE50509[52], GSE61992[53], and GSE99898[54], each containing pre- and on-treatment expression data with targeted BRAF inhibitors attached with clinical outcome, were merged into one combined dataset. Following quantile normalization using dChip[55], the dataset was curated to remove co-occurrent patient samples, leaving a total of 40 individual paired samples. Using Mantel-Cox log rank analysis, *p* = 0.05 for High-to-Low *PPARGC1A* compared to all other samples.

### Animals and human cancer cell lines

Homozygous outbred nude mice (Jackson Lab: Foxn1[nu]/Foxn1[nu] #007850) were used for xenograft tumor experiments under the auspice of Beth Israel Deaconess Medical Center animal facility and

IACUC approved protocols. If experimental end-points were not met prior, the mice were subjected to humane euthanasia using $CO_2$ asphyxiation if body weight declined by more than 20% or tumors exceeded 2000 mm³ (tumor volume = ½ length × width²). All human melanoma cell lines were obtained from the Broad Institute CCLE collection and authenticated using small tandem repeat profiling. All cell lines were maintained in a humidified incubator at 37 °C with 5% $CO_2$, and if not otherwise indicated, in DMEM (Sigma-Aldrich) with 10% FBS, 100 U/ml penicillin, and 100 mg/ml streptomycin.

## Compounds and antibodies

PLX4032 (S1267), EZH2 inhibitor GSK126 (S7061), pitavastatin (S1759), SB273005 (S7540), Defactinib (S7654), trametinib (S2673), GGTI 298 (S7466), CoQ10 (S2398) were purchased from Selleck Chemicals. EPPS (E1894), iodoacetamide (I1149), geranylgeranyl pyrophosphate ammonium salt solution (G6025), farnesol (F203), methylcellulose (M0512) were purchased from Sigma Aldrich. Cell Counting Kit-8 (C0005) was purchased from TargetMol. Geranylgeranyl Pyrophosphate (63330), GLPG0187 (21792), were purchased from Cayman Chemical. Polybrene (SC-134220), (R)-Mevalonic acid lithium salt (SC-505951), Cholesterol (SC-202539), were purchased from Santa Cruz Biotechnology. YM-53601 (NC0701380), Propidium Iodide (P1304MP), Primocin (NC9392943), Geneticin (10121035), Matrigel matrix (CB40234A), One Shot™ CcdB Survival™ 2 T1R Competent Cells (A10460), Gateway BP Clonase II Enzyme mix (11789020), LR Clonase II Enzyme mix (11791020), Pierce™ Cell Surface Biotinylation and Isolation Kit (A44390), Lipofectamine 3000 (L3000015), Terrific Brotch (22711022), RAB6B antibody (PA598909) and N-ACETYL-L-CYSTEINE (A1540914) were purchased from Thermo Fisher Scientific. D-luciferin (E1605) was purchased from Promega. QIAprep Spin Plasmid Miniprep Kit (27106) was purchased from Qiagen. Cilengitide (76325-512), MycoAlert™ Plus Mycoplasma Detection Kits (75860-358), were purchased from VWR.

iScript™ Advanced cDNA Synthesis Kit (1725038) was purchased from Bio-Rad Laboratories. NEB Stable Competent E. coli (High Efficiency) (C30401), Q5 Site-Directed Mutagenesis Kit (E0552) were purchased from New England Biolabs. RIPA buffer (9806 S), Cyclin D1 antibody (92G2), Rabbit mAb (2978 S), Rab11b antibody (2414), Phospho-FAK antibody (Tyr397) (3283), Integrin beta-1 antibody (4706), Integrin alpha-V antibody (4711), Integrin beta-3 antibody (4702), RAB27A antibody (SC-74586) were purchased from Cell Signaling Technology.

Integrin α1 antibody (SC-271034), Rap1A antibody (SC-373968), PGC1α antibody (SC-518025) were purchased from Santa Cruz Biotechnology. H3 (ab1791) antibody, H3K27me3 antibody (ab192985), H3K27ac antibody (ab177178) were purchased from Abcam.

## Cellular adaptation to chronic treatment with BRAF-inhibitor

To obtain cells chronic adapted to BRAF-inhibitor treatment, $1 \times 10^7$ K029A (might be named as K029 or K029AX in other place), A375P, SKMEL3, G361 and SKMEL5 cells were seeded in p150 mm tissue-culture dishes, and subsequently subjected to 1 µM PLX4032 in the media. At indicated time points, resulting cell numbers and *PPARGC1A* RNA levels were measured. As indicated by proliferation in the presence PLX4032, BRAF-inhibitor chronic adapted cells ware achieved. The obtained cells were maintained in the presence of 1 µM PLX4032 and released one week before experiments. The characteristics of BRAF-inhibitor chronic adapted cells are relatively stable as shown in Supplementary Fig. 3n.

## Cell viability assay

Cell viability was measured using Cell Counting Kit-8 (CCK-8, Dojindo) according to the manufacturers' recommendations. Briefly, 10 µl CCK-8 reagent (100 µl medium per well) was added and incubated for 1 h at 37 °C, 5% $CO_2$, and subsequently, absorbance at 450 nm was measured using a FLUOstar Omega microplate reader (BMG Labtech).

## Compound screening

An anti-cancer metabolism-focused compound library (compound names found in supplementary data 1) was bought from TargetMol. Briefly, $1 \times 10^4$ parental and chronic treated BRAF-inhibitor resistant K029A cells were seeded into 96-well plates and the compounds were added 4 h later at the concentration of 10 µM, 3 µM, 1 µM, 0.3 µM and 0.1 µM. After 72 h treatment, cell viability was measured using CCK-8 assay (as described above).

## In vitro migration assay

Transwell chambers were purchased from Corning Life Science. Cells in 0.1 mL of FBS-free medium were seeded into the upper chamber and incubated overnight. The cells numbers seeded per well were $1 \times 10^5$ K029A parental, K029A BRAFi adapted, K029A BRAFi adapted with PGC1α ectopic expression, and parental A375P BRAFi adapted. Growth medium containing 10% FBS was used in the lower chamber as chemoattractant. The remaining cells on the top of the membrane were removed by a cotton swab. The migrated cells were then fixed and stained with 0.2 % crystal violet solution. The membrane attached with migrated cells was placed on a glass slide; total cells from 3 different fields of view were counted under 10× magnifications with a Nikon 80i Upright microscope.

## In vivo metastasis assay

One million ($1 \times 10^6$) luciferase (the lentivirus mediated introduction of luciferase into cells has been described as before[15]) expressing parental and chronic adapted BRAF-inhibitor resistant K029A, A375P, and SKMEL3 cells were tail-vein injected into 6-week-old male nude mice. The chemiluminescent activity and location of the luciferase-expressing tumors were visualized by the Xenogen IVIS-50TM imaging system equipped with an isoflurane (1.5 %) anesthesia system and a temperature-controlled platform subsequent to i.p. injection of 150 mg/kg D-luciferin in PBS buffer.

## Western blot assays

For Western immunoblotting, cells were lysed in RIPA buffer at indicated time points, and the protein concentration was quantified using the DC protein concentration assay (Pierce) before being subjected to gradient 4–12% (30:1) SDS polyacrylamide electrophoresis and subsequently transferred to PVDF membranes. Specifically, MES SDS running buffer was used in gels to detect RAB6B, RAB27A and Cyclin D1, while MOPS SDS running buffer was used for all other Western blot assays. Membranes were blocked, then probed with primary antibodies O/N, and subsequently using secondary antibodies for 1 h at room temperature.

## Plasmid construction, lentiviral generation, and transduction

CRISPR/Cas9 mediated gene knockout was performed using the GeCKO system, where pLentiCRISPRv2 (Addgene, 98290) was digested with BbsI enzyme and pre-annealed 5′-end phosphorylated sgDNA sequences (the sgRNA sequence can be found in supplementary data 2) inserted using Quick Ligase (New England Biolabs), and subsequently transformed into Stabl3™ E. coli. Resulting plasmid inserts were verified by sequencing (Genewiz). Plasmids pLV-shRBA6B-Bsd, pLV-shRAB27A-Puro, pLV-CBh-hRAB27A-Bsd, and pLV- CBh-hRAB6B-Puro were purchased from Vector Builder.

The Gateway® Technology was used to generate lentiviral expression constructs for MITF, RAB32, RAB38 and RAB7A. RNA was obtained from K029A cells by TRIzol (Thermo Fisher Scientific) and cDNA was synthesized by the high-capacity cDNA Reverse Transcription Kit (Life Technologies, 4368813). ORF fragments were amplified

by Q5® High-Fidelity DNA Polymerases (NEB, M0419) with the corresponding pairs of primers (in supplementary data 2). The obtained ORF fragments were introduced into pDONR223 by BP Clonase™ Enzyme Mix (ThermoFisher, 11789013) and into pLX301 and pLX304 by LR Clonase™ II Enzyme mix (ThermoFisher, 11791020). The constructs pLX304-LacZ, pLX301-LacZ, pLX304-PGC1α and pLX301-PGC1α have been described previously[6]. Replication-defective lentiviral supernatants were generated using transfection of 600 ng psPAX2 (Addgene, #12260), 300 ng pMD2.G (Addgene, #12259) and 900 ng lentiviral plasmid backbone using PolyFect (Qiagen, 301105) into HEK293t cells in 6-well format according to the manufacturer's instructions. Supernatants collected twice (at 48 and 72 h) post-transfection, and filtered through a 0.45-μm filter, then used to transduce cells in the presence of 8 μg/ml polybrene. The transduced cells were then selected with 2 μg/ml of puromycin or 8 μg/ml blasticidin for 4 days and then cultured without antibiotics for at least 7days prior to use in experiments.

## Quantitative real-time PCR (qPCR)
RNA was isolated with Trizol (Invitrogen, 15596-026) and the Zymo-Spin Direct-zol RNA Kit (Zymo Research, R2050). 1 μg of RNA was used to generate cDNA with a High-Capacity cDNA Reverse Transcription Kit (Applied Biosystems, 4368813) following the manufacturer's protocol. For gene expression analysis, cDNA samples were mixed with Sybr Green quantitative PCR master mix (Applied Biosystems, 4309155) and were analyzed by a CFX 384 Real-Time system (Bio-Rad). All primers sequences can be found in supplementary data 2.

## Chromatin immunoprecipitation (ChIP)
ChIP was performed using the ChIP kit (Millipore) with slight modification. Following sonication, nuclear lysates were precleared with protein A/G-Dynabeads (Invitrogen) for 1 h. Equal amounts of precleared lysates were incubated with IgG or specific antibodies (H3K27me3 from Abcam) overnight, followed by precipitation with protein A/G-Dynabeads for 2 h. To quantify the promoter occupancy, qPCR was performed.

## Tumor xenograft studies using melanoma cell lines
$1 \times 10^6$ of melanoma cells ($1 \times 10^7$ cells of K029A parental cells) were injected subcutaneously into the right flank side of nu/nu mice, and when tumors reached approx. 150 mm$^3$ (tumor volume = ½ length × width$^2$), cohorts ($n = 5$ for K029A, $n = 1 0$ for A375P and $n = 5$ for SKMEL3) were treated (i.p., b.i.d.) with either pitavastatin (1 mg/kg) or PBS as vehicle control. Tumor sizes were longitudinally measured and followed until a mouse within one cohort had a tumor >1500 mm$^3$. No mice exhibited severe loss of body weight (>15%) or any evidence of ulceration within these experiments. Statistical analyses were based on the tumor volumes at the final time points.

## Isolation of membrane-associated proteins
Cells in p100 mm tissue culture dishes were washed with PBS twice and then detached by trypsin treatment. Cell pellets were resuspended in lysis buffer (10 mM HEPES, pH 7.4, 250 mM sucrose, and proteinase inhibitor) and then sonicated. Cell debris and nuclei were removed by two subsequent centrifugations (2500 rpm, 5 min, at 4 °C) to clear the supernatant. Membrane fractions were then collected by centrifugation (200,000 × g, 30 min, at 4 °C) and the resulting membrane pellets were resuspended in RIPA buffer with proteinase inhibitors and subjected to Western blot analyses.

## Cell surface protein biotinylation and capture
Proteins localized at the plasma membrane cell surface were captured by Cell Surface Biotinylation and Isolation Kit (Pierce) according to the manufacturer's recommendations. Briefly, cells in p150 mm tissue culture plates were treated with pitavastatin (1 μM; 18 h) and and

washed twice with PBS. Cell surface protein was labeled with EZ-Link sulfo-NHS-SS-biotin and then captured with NeutrAvidin Agarose. The bound protein was then eluted and subjected to mass spectrometry-based proteomics and Western blot analyses.

## Peptide labeling with tandem mass tags
Proteomics was performed as essentially as previously described[26]. Briefly, parental or chronic treated BRAF inhibitor resistant K029A cells ($1 \times 10^7$ cells) were treated with pitavastatin (1 μM; 24 h), and subsequently trypsin treated and collected by centrifugation. Cell pellets were solubilized in the lysis buffer (8 M Urea/100 mM HEPES pH8.5) reduced with 5 mM TCEP at 60 °C for 30 min, alkylated with 14 mM iodoacetamide for 45 min in dark, and precipitated using methanol chloroform at the ratio as (sample: methanol: chloroform: H$_2$O; 1: 3: 1: 2.5). Samples were centrifuged (4000 g; 10 min) and pellets were washed three times with cold methanol. Extracted proteins were resuspended in 200 mM EPPS buffer and digested with Lys-C (1:100 protease-to-protein ratio) (3 h; 37 °C) following trypsin digestion (1:100 protease-to-protein ratio) at 37 °C overnight. Peptides were quantified by microBCA reagent (Thermo Fisher Scientific) and TMT labeled (6–8 M excess). Formic acid was added to a final concentration of 1% and peptides were clean up using a 50 mg Seppak column. Eluted peptides were dried down and resuspended in 5% ACN/5% formic acid buffer. Ratios were checked by HPLC, and samples fractionated. Fractions were resuspended in 1% formic acid and cleaned up using C18-membrane stage tips. Samples were dried down and MS analysis was performed as previously described.

## Cell attachment
Parental and chronic treated BRAF-inhibitor resistant K029A and A375P cells were pretreated with pitavastatin (1 μM; 18 h), resuspended in growth medium, and subsequently seeded ($1 \times 10^6$ cells/well in 6-well format) to allow for attachment during 2 h, and then washed to remove non-adhering cells prior to fixation and crystal violet staining (0.2%).

## Detachment-induced cell death (anoikis) assays
Cells (50,000 cells/ml/35 mm plate) were suspended in growth media containing 0.5% methylcellulose to prevent cell-cell interactions and treated with NAC (5 mM) or pitavastatin (1 μM), and/or with 5% percent matrigel. After the indicated time point, cells were spun out, washed twice, and then counted in the presence of trypan blue (0.2%) staining to assess frequencies of live-to-dead cells.

## FACS-based cell cycle analyses
Cells suspended in PBS were fixed using ice-cold 70% ethanol and then treated with 0.2 mg/ml RNase for 1 h at 37 °C subsequently resuspended in PBS with PI (2 μg/ml). PI stained cells were then analyzed on BD FACSCanto II flow cytometer.

## Statistical analysis and reproducibility
All measurements were biological replicate samples (different experimental days for same cell line). All western blot analyses were performed in three biological independent experiments. In general, for two experimental comparisons, unpaired two-tailed Student's t-test was used. Statistical significance is represented by asterisks corresponding to $*p < 0.05$, $**p < 0.01$, and $***p < 0.001$. GraphPad Prism software (version 8.4.0) was used to generate graphs and perform statistical analyses. Microsoft Excel was used for analysis of small-molecule screen data and proteomics. DAVID Bioinformatics Resources 6.8 was used for gene ontology analysis.

## Reporting summary
Further information on research design is available in the Nature Portfolio Reporting Summary linked to this article.

## Data availability

The authors declare that the data supporting the findings of this study are available within the paper and its Supplementary Information files and will be available from the corresponding author upon reasonable request. The mass spectrometric-based proteome data generated in this study have been deposited in the ProteomeXchange Consortium - PRIDE repository database under accession code PXD041952. Source data is available as a source data file. Source data are provided with this paper.

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

## Acknowledgements
We thank members of the Puigserver laboratory for helpful discussions regarding this project. This work was in part supported by the Claudia Adams Barr program in Cancer Research (to P.P.), Dana-Farber Cancer Institute internal funds (to P.P.), Beyond the Sun Drenched Skies philanthropic fund (to H.R.W.), NIH R01CA181217 (to P.P.), Friends of Dana-Farber award (to J.L.), CRI Fellowship CRI4166 (to J.L.) and an AACR-Merck Immunooncology Research Fellowship 22-40-68-YU (to D.Y).

## Author contributions
C.L. and P.P. initially conceived the premise of this work, and H.R.W provided guidance and clinical correlative analyses. J.L., D.Y., and C.L. carried out the majority of experiments and was supported by C.B.; M.J. and S.G. performed mass spectrometric-based proteome analyses. J.L., H.R.W. and P.P. largely analyzed the data and wrote the manuscript.

## Competing interests
The authors declare no competing interests.
