## [Peer Review File · Nature Communications]

Epigenetic suppression of PGC1 α (*PPARGC1A*) causes collateral sensitivity to HMGCR-inhibitors within BRAF-treatment resistant melanomasREVIEWER COMMENTS

Reviewer #1 (Remarks to the Author):

In this manuscript Liang et al show that a particular subset melanoma patients that initially express high levels of PGC1a- and upon BRAF inhibitor treatment -downregulate PGC1a and have worse overall survival. Authors use melanoma cell lines to in vitro recapitulate this scenario (initially high PGC1a expression that becomes downregulated when cells are resistant to BRAFi). They go on to show that PGC1a is epigenetically downregulated in this subset of BRAFi resistant cells. Moreover, these cells are more sensitive to statins, and authors suggest that this is a collateral metabolic vulnerability that could be exploited in the clinic.

Although the need to find combinatorial treatments to overcome resistance to current therapies in melanoma is undeniable and that the idea behind this manuscript has good potential, the paper as currently conceived leaves important gaps to be addressed. In particular, the more mechanistic aspects of the study have not been proven and several conceptual gaps in experimentation do not support the proposed conclusions. Specific points are listed below.

1- Authors show that chronic BRAF inhibitor treatment drives epigenetic downregulation of PGC1a in a subset of melanoma cells. How is BRAF inhibitor treatment inducing these changes in methylation and acetylation on PGC1a promoter?

Authors should show similar data with another cell line.

2- To find vulnerabilities within this subset of melanoma cells, authors use a metabolism-focused compound library comparing growth inhibition in drug naïve versus BRAFi-resistant cells. What is the exact rationale behind using this library of compounds?

3- Authors claim increased sensitivity to HMGCR inhibitors in a subset of BRAF inhibitor resistant melanomas throughout but only one compound is tested, Pitavastatin (other than in the initial screen). Key findings should be proven with other drugs to make such statement.

4- Authors show that BRAFi resistant cells that downregulate PGC1a are more sensitive to statins while BRAFi resistant cells that maintain PGC1a levels are not. What makes this subset of melanomas more sensitive to statins? Would silencing of PGC1a induce sensitivity to statins in the other subset of BRAFi resistant cells? In the same direction, do cells with different levels of PGC1a have different sensitivity to statins (naïve to BRAF inhibitor treatment). If statins prevent prenylation of RAB proteins, impairing their function, why are parental cells not sensitive to the effects observed in BRAFi resistant cells in terms of cell adhesion and p-FAK?

5- Authors use CCK8 assay to measure cell viability in the drug screen and also to compare sensitivity to statins in BRAFi resistant vs parental cells (Fig 2). Given the data from figure 4 where it is shown that pitavastatin decreases membrane integrin expression and adhesion in BRAFi resistant cells, authors should demonstrate that effects observed using CCK8 assay are due to changes in viability rather than just changes in cell number due to defective adhesion.

6- Data showing a link between MITF, PGC1a, RAB proteins and sensitivity to pitavastatin is not very conclusive. Authors show that MITF regulates PGC1a expression and in turn PGC1a regulates RAB expression, which according to authors drive changes in sensitivity to pitavastatin. Indeed, disruption of either MITF or PGC1a using CRISPR/Cas9 sensitizes cells to pitavastatin. However, ectopic expression of PGC1a in MITF depleted cells is not able to rescue the phenotype. Authors suggest that "appropriate expression of each PGC1a and MITF control HMGCR-inhibitor sensitivity" but this explanation is not convincing. Is MITF regulating RAB expression independently from PGC1a?

7- On the one hand, PGC1a regulates RAB expression, which is responsible for increased sensitivity to statins in BRAFi resistant cells. On the other hand, authors show that GGPP is able to rescue the effect induced by statins in BRAFi resistant cells, while other metabolites from the mevalonate pathway aren't, suggesting that impairment of RAB prenylation, and therefore traffic to the membrane, is driving sensitivity to statins. Given that in the subset of melanomas under study PGC1a levels are very low-same as RAB levels, how is it possible to rescue the effect of statins to the same extent using either RAB6B/27A overexpression or GGPP supplementation? Also, are RAB6B/27A WT ectopically expressed already prenylated and therefore not sensitive to lack of GGPP due to statins treatment?

8- Authors suggest that integrin mRNA expression is modulated by PGC1a and by RAB6B/27A. They show high mRNA expression of integrins in low PGC1a expressing cells (both parental cells with sgPGC1a and BRAFi resistant cells) and high mRNA expression of integrins in BRAFi resistant cells with ectopic expression of RAB6B/27A. How is integrin mRNA expression regulated by PGC1a?

And by RAB6B/27A? How is this linked to statins sensitivity?

Minor comments:

- 1- Tumours from K029A-R-RAB6B/27AWT and MT grow faster than K029A-R-LacZ (Fig 3h), how do authors justify these differences?
- 2- Mass spectrometry proteomic data shows a decrease in several RABs but in figure 3 authors only focus on RAB6B and RAB27A. Authors should justify why they focus only in these two.
- 3- Authors should explain the rationale behind the combinations used in Extended Data Fig. 7a, b)
- 4- Key findings should be repeated with another cell line
- 5- Authors should double check figure citations throughout the manuscript, there are some mistakes. Also, figures should be arranged in the order the panels are cited in the text.
- 6- Authors should indicate the N of each experiment.
- 7- Authors should make sure the bibliography is adequate. For instance in this sentence the references cited are not relevant "While there are preclinical studies suggesting that HMGCR inhibitors alone or in combination with other drugs effectively inhibit melanoma cell growth 22,27".

Reviewer #2 (Remarks to the Author):

In this manuscript, Liang et al identify a subset of BRAF inhibitor-resistant melanomas that emerge during chronic drug treatment and show that they are associated with decreased levels PPARGC1A, because of increased levels of H3K27me3 and reduced levels of H3K27ac marks across the PPARGC1A promoter region. PPARGC1A silencing in these tumors are associated with metastatic traits but leads to sensitivity to HMGCR-inhibitor treatments. They also show that suppression of PPARGC1A results in lower RAB6B and RAB27A levels that in response to HMGCR-inhibitor treatments reduces cell surface trafficking of integrin and downstream FAK signaling (which are elevated in BRAF inhibitor-resistant cells).

The topic of drug resistance in melanoma and the challenge to overcome such resistance is clinically significant. However, there are major limitations that diminishes the enthusiasm for this work.

1 - In the introduction of the manuscript, the authors only review the impact of MAPK/ERK pathway PI3K/AKT pathway as known mechanisms of BRAF inhibitor resistance in melanoma. Then, they motivate their studies, based on the idea that "It is currently not entirely known whether targeted BRAF-drug resistance is widespread, or only found pre-existing within a small population of cancer cells. Importantly, it is not understood whether there are any underlying mechanisms that can be used to inform treatment resistance, and whether co-therapeutic strategies can prevent or target emerging resistance." However, there are many impactful papers published in the past 5 years that have investigated the role of epigenetic adaptation and differentiation state plasticity on MAPK inhibitor resistance. The connection of PPARGC1A and the differentiated (MITF^{High}) state has also been previously discussed. The authors should improve the introduction to the previous studies and specifically review the more recent literature of the BRAF/MEK inhibitor resistance in melanoma that are more relevant to their work.

2- Most studies in this manuscript are performed in one or two cell lines. This raises questions and concerns about the generalizability of the findings. This is especially important, given that diverse cell lines do behave differently with respect to induced changes in the expression of differentiation markers (e.g., MITF) and PPARGC1A following drug resistance. Some cell lines show increased expression and in only some cell lines (that are selected for this study), PPARGC1A levels drop following MAPK inhibitor treatment.

3- The authors focus their study of chronic BRAF inhibitor resistance on PPARGC1A, based on the finding that the PPARGC1A levels drop significantly as shown in Fig. 1c, 1d. However, the drop of PPARGC1A levels happen much earlier than the emergence of proliferating drug-resistant cells. So, does the PPARGC1A depletion lead to drug tolerance at earlier timepoints? If so, shouldn't we expect synergy between pitavastatin and BRAF/MEK inhibitors at least in cell culture?

4- The authors conclude that "epigenetic" silencing of PPARGC1A drives metastatic traits and leads

to collateral sensitivity to HMGCR-inhibitor treatment. This is based on the measurements of two histone marks H3K27me3 and K3K27ac and their changes following chronic drug resistance (Fig. 1g). However, as mentioned above, PPARGC1A silencing begins as early as a few days post-drug treatment in K029 cells (Fig. 1c). So, do these histone marks change within the same time scale (i.e., after a few days)?

5- What makes such epigenetic changes possible in some cell lines but not in others? Are the epigenetic modifications that result from H3K27me3 and K3K27ac changes affect only PPARGC1A, or a larger pattern of gene expression associated with the drug-resistance state?

6- In vivo studies presented in Fig. 2g-2k are done in various lengths from 9 days to >20 days. Since the authors compare these data to make conclusions about the underlying conditions for pitavastatin efficacy, it is surprising to see that they have not performed these experiments for the same durations. It is clear, for example, that 10 days of treatment with pitavastatin in Fig. 2h does not provide significant efficacy in A375P-R tumors and treatments need to continue for >20 days for the drug to be effective. So, why did the authors terminate the experiment on A375P tumors after only 9 days (Fig. 2i)?

Reviewer #3 (Remarks to the Author):

The role of PGC1a in tumorigenesis has been addressed in many previous publications, including in several studies on melanoma. In particular, Puigserver and his group have shown before that PGC1a is essential for survival of a subset of melanoma cells and that it regulates integrin expression and invasiveness / metastasis formation. In the present manuscript, Puigserver and colleagues extend these findings and reveal that, in a subset of melanoma cells emerging upon chronic BRAFi treatment, PGC1a expression gets suppressed, which is associated with more aggressive (invasive and metastatic) features in human cell lines. Using a metabolism-focused compound library screen, the authors identify BRAF-inhibitor resistant melanoma cells with silenced PPARGC1A to be sensitive to HMGCR-inhibitors in vitro and in vivo. Mechanistically, the authors show that PGC1a regulates RAB6B and RAB27A, which are important for sensitizing PPARGC1A low BRAFi-resistant cells to HMGCR-inhibitors (such as pitavastatin) also in vivo. Further, pitavastatin reduced membrane association of RAB6B and RAB27A affecting cell surface integrin receptor association and downstream FAK activation, leading to reduced cell growth.

This is an interesting paper that convincingly links PGC1a to targeted therapy resistance in a subset of melanoma cells. However, some important issues need to be addressed before publication.

Major points:

- Two out of four BRAFV600E-mutant melanoma cell lines show reduced PPARGC1A mRNA expression in BRAFi resistant cells (Ext.Fig.1b). However, the authors do not show whether BRAFi-resistant cell lines with silenced PPARGC1A (K029A, A375P) are indeed more aggressive than BRAFi-resistant cells displaying increased PPARGC1A (G361, SKMEL5), as predicted from the analysis of patient data (Fig. 1b). The authors should extend the data shown in Fig. 1e by investigating these additional cell lines in vivo.

- On the same token, the authors conclude from the above-mentioned experiments, that their main finding ("Epigenetic suppression of PGC1a (PPARGC1A) causes collateral sensitivity to HMGCR inhibitors within BRAF-treatment resistant melanomas" (Title)) only applies to a subset of melanoma cells. To substantiate this important statement, the authors should use at least three cell lines representing this subset of BRAFi-resistant melanomas for their key experiments. Many experiments have been done with just one cell line, some with two. Furthermore, the authors should show that in cell lines not representing this specific BRAF-resistant subset (such as G361, SKMEL5), PPARGC1A is not epigenetically suppressed by EZH2 and that the proposed downstream effectors pFAK, RAB6B and RAB27A are differentially expressed in K029A, A375P, etc., as opposed to G361, SKMEL5.

- Low PPARGC1A expression has been associated with emergence of resistant cells, but decrease

in PPARGC1A expression occurs way before resistant cells emerge (Fig. 1c,d). This rather speaks against a direct connection between these two processes. How do the authors explain this? Is, for instance, the regulation of RAB6B and RAB27A downstream of PGC1a indirect and delayed compared to the downregulation of PPARGC1A upon BRAFi treatment?

- The authors propose a link between epigenetic regulation of PPARGC1A and its role in resistance formation. Does the kinetics of H3K27me3 occupancy at the PPARGC1A promoter follow its expression during BRAFi treatment? DZNep is not specific for EZH2, the use of a more specific inhibitor would be preferable. Moreover, would HDACi treatment interfere with resistance formation? Such experiments are relevant because the authors emphasize the role of epigenetic suppression of PGC1a in the processes described here.
- To strengthen the functional link between PGC1a and RAB27A and RAB6B, rescue of pitavastatin-treated cells by overexpression of RAB27A and RAB6B (Fig. 3, Extended Data Fig. 6) should also be done on a sg-PPARGC1A background.
- Related to Figure 4, the changes in pFAK levels should be shown in vivo (in settings as presented, e.g., in Fig. 3h). Furthermore, Marin-Bejar et al (Cancer Cell 2021) have previously reported a functional association between FAK expression and resistance formation to targeted therapy. This should be discussed.
- The relationship between PGC1a and MITF (Ext. Data Fig. 5) could be strengthened and should be further discussed in the context of existing literature. It is intriguing that MITF and PPARGC1A, respectively, cannot rescue each other in the experiments presented in Extended Data Fig. 5d,e. How do the authors explain this, given the close relationship between MITF and PPARGC1A? The statement on p.5 "that appropriate expression of each PGC1a and MITF controls HMGCR-inhibitor sensitivity, ... presumably by regulating RAB6B and RAB27A levels" could be experimentally addressed.
- In general, the methods are poorly described. For instance, the authors should explain in the method section how they analyzed the publicly available datasets shown in Fig. 1a, b. Further, they should specify how long they maintained the specific cell lines under BRAFi treatment, and whether the phenotype remained stable over time – especially, when the cells were modified.
- Statistics are insufficiently described. Almost all figures are displayed as bar graphs without ever mentioning the number of replicates shown and without displaying the actual data points. Furthermore, there is no information on how many times representative western blots have been repeated. This must be specified.
- The data presented in Fig. 4d need to be quantified.

Minor points:

- In Fig. 1f, the authors show that overexpression of PPARGC1A abolishes cell migration in BRAFi-resistant K209A cells, indicating that the migratory effect is dependent on PGC1a. The authors should show validation of the overexpression in this experiment.
- In the figure legend to Fig. 1g and Fig. 3j, the authors named the cell line K029S. Is this a different cell line than K029A?
- The figure order is confusing, making the paper (which is anyway written in a very compact manner) difficult to read at certain places. Authors should rearrange figures (especially Extended Data Figures 3 and 4). Further, they should cite the figures in an alphabetical order in the text and mention all the figures in the text (they did not refer to Fig. 2b in the text, for instance).
- In Fig. 3j, the authors show that silencing of RAB27A and RAB6B individually sensitizes K029S (A?) cells to pitavastatin. The authors claim that double silencing further increases statin sensitivity, however, this is not obvious from the graphs presented.
- On p.5, "we searched our mass spectrometric-based proteomic data which evidenced a decreased amount of RAB6B and RAB27A in chronic treated BRAF-inhibitor resistant K029A cells compared to their parental counterparts" should refer to Extended Data Fig. 1e, not Extended Data Fig. 1d.

Reviewer #4 (Remarks to the Author):

The authors determine that epigenetic suppression of PGC1alpha is associated with an aggressive subset of BRAF-inhibitor resistant melanoma. They go on to identify that tumor cell lines with repressed PGC1alpha expression are particularly sensitive to statins. Mechanistically they invoke that statin exposure leads to depletion of GGPP pools, which results in the decrease in the isoprenylation of RAB proteins. In particular, they argue that RAB6B and RAB27A play a key role in the growth and survival of this subset of melanoma. They further suggest that there is a decrease in integrin complexes and reduced FAK activity. The authors have made a substantial contribution; however the direction of research is not always rationalized and data interpretation is not always considered in full. Issues associated with attention to detail are also problematic.

1. The entire manuscript is performed on two parental melanoma cell lines (K029A, A375P) and their BRAF-inhibitor resistant counterparts that demonstrate a 2- to 4-fold decrease in mRNA expression of PPARGC1A, a transcriptional coactivator and master regulator of mitochondrial biogenesis. The results would be significantly strengthened with additional data in relevant melanoma patient-derived model systems, such as organoids or PDXs. Evaluation in immune-competent melanoma models would also add weight. Results in these types of relevant models would also support the authors' conclusion that the preclinical data from this manuscript supports a clinical trial. Without these additional models, the results are quite limited as they largely focus on K029A cells.
2. How does reduced PPARGC1A expression lead to enrichment in plasma membrane, ECM and integrin proteins, as well as a decrease in RAB protein expression, as shown in Ext Fig 1e? Is this at the level of transcription? RNAseq and ChIP-seq data would provide mechanistic insight.
3. Why is the K029A control not shown (Fig 2g)? Experiments conducted at the same time should be displayed on the same graph (e.g. Fig 2h and 2i; Fig 2j and 2k).
4. Ext Fig 4: The authors have not provided data to show that the metabolites were taken up by the cell. Thus, either this data needs to be shown or this caveat must be included in the text as part of data interpretation.
5. Fig 3c and 3d: The authors state that melanoma biopsies (TCGA: SKCM) and cell lines (Broad: CCLE21Q1) datasets show RAB6B and RAB27A mRNA expression was highly correlated with PPARGC1A mRNA levels. However, other RABS (RAB38, RAB7A, RAB32) are also highly correlated, yet they are dismissed. Other RABS are also downregulated in BRAF inhibitor resistant cells compared to parental (Ext Fig 1e). The authors need to show the functional role of these RABS. This is important because neither 6B or 27A was functionally important on its own, suggesting a class effect. It is likely that the loss of isoprenylation of several RAB proteins is contributing to pitavastatin activity. The rationale for focusing only on two RAB proteins is not justified. If it is a class effect of all isoprenylated proteins, including several RABS, then that is the message that needs to be delivered. Emphasizing the two RAB proteins alone as holding key functionality is not supported by the data as the authors have not shown that the other RAB proteins don't also contribute to the effect of pitavastatin exposure.
6. Are 6B and 27A preferentially affected at the level of isoprenylation in response to pitavastatin? Evaluating levels of prenylated and unprenylated forms of several RAB protein levels in the presence or absence of pitavastatin treatment would determine if there was preferential loss in the isoprenylation of specific RABS, which may explain why these RABS (6B, 27A) are functionally critical. Mass spec or other types of isoprenylation assays may be useful here.
7. Information regarding RAB6B and RAB27A need to be provided so the reader might understand better why these two RABS are particularly functionally important for the growth and survival of these cells. It is hard to understand why ectopic RAB6B is so important in combination with RAB27A in terms of the rescue effect. This is particularly difficult to understand given that RAB6B appears difficult to ectopically express in the presence of RAB27A (e.g. Ext Fig 6d), or visa versa (Ext Fig 8c). Was RAB6B ectopic expression more robust at the protein level on its own and/or when combined with other RABS (e.g. 7A, 32, 38; Ext Fig 7 only shows mRNA). As written the manuscript is difficult to understand because the authors don't provide rationale for why they are getting the results they see with one RAB vs another. Perhaps providing more background would be instructive. Are these RABS essential to all melanoma cell growth and survival or for only this subset of melanoma? Etc. Dialogue regarding the results would be helpful in terms of rationalizing data interpretation.
8. To help resolve and provide clarity, the authors need to evaluate whether GGTase inhibitors

phenocopy pitavastatin. Are BRAFi-R cells also sensitive to geranylgeranyl transferase inhibitors?
9. Fig 3e, 3f, Ext Fig 5a, Ext Fig 5b: i) What happens to the mRNA expression of the other RAB proteins under these conditions (i.e. CRISPR PGC1alpha, CRISPR MITF, overexpression of PGC1alpha)? Are other RABs similarly regulated as RAB27A and RAB6B? ii) What is the relative growth rate of these cells? Could the effects on expression of the RABs be a reflection of relative cell proliferation/cell death under these various conditions? For example, is the change in RAB expression due to changes in PGC1alpha expression (cause) or due to an indirect effect on cell proliferation (consequence). Knowing the cell proliferation/cell death of these populations will help to determine whether the effect is a cause or consequence.

10. Fig 3g: Does ectopic expression of additional RAB proteins further the rescue effect seen with ectopic expression of RAB6B+RAB27A? Only RAB6B and the other RAB proteins was interrogated.

11. Fig 4a: The authors indicate HMGCR as up-regulated within the figure, but don't explain this in the text as an expected change due to the well-established feedback response to statin exposure that leads to the induction of several mevalonate pathway genes, including HMGCR. It is surprising that other genes in this metabolic pathway don't appear to be up-regulated, yet they (like HMGCR) should serve as positive controls. Loss of this feedback loop is associated with increased sensitivity to statin exposure (for review see PMID 32887721). If these mevalonate pathway genes are not up-regulated, then this loss of feedback may also be contributing to pitavastatin sensitivity. These mevalonate pathway genes need to be highlighted in the volcano plots as positive controls, e.g. HMGCS1, LDHA, INSIG, etc.

12. Fig 4a, 4b: Are the changes in BRAFi-R cells due to increased genomic instability compared to parental cells? Would similar changes be seen in response to any perturbation? Is the induction of DNA damage required for pitavastatin-induced cell death? Statins do not directly damage DNA. The authors need to frame the results into context and provide the reader guidance on how to interpret these data, rather than just present the results.

13. Fig 4c: As the authors see evidence of Parp cleavage (Fig Ext 8C) it is important to determine what fraction of cells are undergoing cell death by gating on sub-G1 in these flow cytometry experiments. Also, it is the industry standard to show 3 independent measures of apoptosis, e.g. TUNEL, Annexin V staining, TMRE as a measure of mitochondrial potential, etc.

14. Fig 4e. "This approach revealed that pitavastatin treatment reduced cell surface localization of a large number of integrins within the chronic treated BRAF-inhibitor resistant cells, which were correspondingly retained by RAB6B and RAB27A over-expression (Fig. 4e, Extended Data Fig. 10b)." However, the data show that pitavastatin reduced about half of all integrins (Fig 4e). Indeed, the two used for further validation are decreased (A5) and increased (B1) in this figure. The scale is shallow, which could be contributing to the display being misleading? This needs to be addressed either in the text and/or the data display.

15. The introduction of NAC into the experimental rationale is missing. It comes out of the blue. Isn't NAC really serving as a positive control rather than a discovery aspect of the work? Rationale is needed for why suddenly an anti-oxidant is being evaluated. It is important to remember the broad readership of Nature Communications.

16. Also, why is cell survival under de-attachment growth conditions (anoikis) being measured? Rationale again not provided. What about measuring classic apoptosis under regular growth conditions on adherent plates? The latter half of Fig 4 seems distant to the main thrust (see title) of the manuscript. Is this tangent necessary? If so, rationale is required. Even the authors state, "These data indicates that resistance to anoikis driven by elevated integrin receptor expression and FAK signaling is functionally separable from the collateral sensitivity to HMGCR inhibitors seen in chronic treated BRAF-inhibitor resistant melanoma cells with silenced PGC1 expression." Perhaps a working model figure would help to add clarity.

17. Abstract: "PGC1alpha suppression drives transcriptional downregulation of RAB6B and RAB27A levels..." Changes in mRNA expression does not necessarily mean that PGC1alpha is regulating these genes at the level of gene transcription. If so, the authors need to show how. What is the mechanism?

18. The experimental datasets were not provided as supplementary tables, e.g. Fig 4a, 4e, Ext Fig 1e, Ext Fig 2a, Ext Fig 10b. This is important as this is the primary data from which further validation is shown throughout the manuscript.

Minor:

1. As ITGB5 is further studied, it would be appropriate to highlight and label the data point corresponding to this protein in Ext Fig 1e showing MS proteome data of BRAFi-R vs parental cells.

2. It is important to mention how the drug treatment of pitavastatin used in mouse experiments (1mg/kg; see Fig 2g-k) relates to the dosing of pitavastatin used to lower serum cholesterol in humans. It is also important to relate this dose to that used for in vitro studies, in terms of micromolar equivalents. This provides the reader rationale as to why the doses used in this manuscript were chosen and how they relate to those used clinically and to levels achievable in human plasma.
3. Background information regarding statins and melanoma are notably missing from the manuscript. This is not the first report showing melanoma sensitivity to statins. In addition, statin-use has been shown to be associated with better outcome in melanoma, a concept that should be incorporated into the manuscript.
4. Legends and text are incomplete, which forces the reader to turn to materials and methods to better understand the experiment being performed. Provide clear description of all experiments in the text and/or legend so reader can easily follow along. E.g. Fig 1e; e.g. providing dose and time of pitavastatin used throughout all experiments, along with rationale for these choices.
5. Several examples of figure labels that are difficult to read, missing or misaligned with the data, including Fig 1f, Ext Fig1c, Ext Fig 6d, can't see protein labels within Fig 4a volcano plot (right); Ext Fig 10b, Ext Fig 8b, and Ext Fig 10a.
6. Figures are not in order in the text, e.g. change Ext Fig 3d, 3e to 3b, 3c to have figures in chronological order in text; similarly Ext Fig 4d, and Ext Fig 6c.
7. Can't read labels highlighted in blue, Ext Fig 4d. This figure should be Ext Fig 4a as the mevalonate pathway is introduced in the text prior to the data with the mevalonate metabolites.
8. This statement "Since GGPP is mainly used to prenylated protein substrates..." is not supported by references that provide data showing that the majority of GGPP is used for protein isoprenylation compared to other GGPP end-products such as CoQ and dolichol. I don't know of such reference(s). Thus, this statement may be unfounded and need to be rewritten.
9. Authors 'need to check this': see text "Tumor xenograft studies using melanoma cell lines (NEED TO CHECK THIS!!!!)"
10. Text states A375P and K029A, yet data pertaining only to the latter is shown. Show A375P data too. "In agreement, chronic treated BRAF-inhibitor resistant A375P and K029A melanoma cells exhibited reduced expression of these two RAB genes (Fig. 3e)."
11. As MITF is investigated in the results section, the role of MITF in melanoma and its relationship with PGC1alpha needs to be mentioned in the introduction as Nature Communications has a broad readership that may not be familiar with MITF-PGC1alpha. Similarly, an intro on the RAB family of proteins would also help support readers, which is particularly important as so much of the work focuses on the RABS. Same could be said for integrins, FAK, etc.
12. Statistical analysis of Fig 3j is needed. Are these differences statistically significant?
13. Ext Fig 6b: what are the relative growth kinetics of KO29A-R-RAB6B/27A WT vs MT? Is pitavastatin just targeting the faster growing tumor?
14. Ext Fig 7b: which cells are being evaluated here?

ANSWERS TO REVIEWERS' COMMENTS

We sincerely appreciate the reviewers' salient critique and thank them for time invested to help improve our manuscript. To address the concerns raised, we have performed multiple additional experiments and appropriately revised the text. We certainly feel that this revised manuscript has been strengthened to detail and scope, whereby the conclusions are now all firmly supported by experimental evidence. Please find below point-by-point responses to each reviewer's individual points of critique:

Reviewer #1:

In this manuscript Liang et al show that a particular subset melanoma patient that initially express high levels of PGC1 α and upon BRAF inhibitor treatment - downregulate PGC1 α and have worse overall survival. Authors use melanoma cell lines to in vitro recapitulate this scenario (initially high PGC1 α expression that becomes downregulated when cells are resistant to BRAFi). They go on to show that PGC1 α is epigenetically downregulated in this subset of BRAFi resistant cells. Moreover, these cells are more sensitive to statins, and authors suggest that this is a collateral metabolic vulnerability that could be exploited in the clinic.

Although the need to find combinatorial treatments to overcome resistance to current therapies in melanoma is undeniable and that the idea behind this manuscript has good potential, the paper as currently conceived leaves important gaps to be addressed. In particular, the more mechanistic aspects of the study have not been proven and several conceptual gaps in experimentation do not support the proposed conclusions. Specific points are listed below.

We thank the reviewer for the summary that captures the essence of our manuscript. Regarding the general critique, we have addressed mechanistic aspects and conceptual gaps in the revised manuscript (see below for the specific responses).

1- Authors show that chronic BRAF inhibitor treatment drives epigenetic downregulation of PGC1 α in a subset of melanoma cells. How is BRAF inhibitor treatment inducing these changes in methylation and acetylation on PGC1 α promoter?

We recently published that expression of PPARGC1A (encoding PGC1 α) is suppressed by histone H3K27 trimethylation within the PPARGC1A promoter region, which could be modulated by inhibiting or suppressing EZH2 function¹. We now provide data suggesting that also in the context of chronic treatment with BRAF-inhibitor, EZH2 function controls PPARGC1A silencing. Specifically, **i**) in a comparison between parental (treatment naïve) and chronic BRAF-inhibitor treated (resistant), we find that EZH2 mRNA transcript levels are consistently increased across each of the three melanoma models, A375P, K029A and SKMEL3, but there was no change in the SKMEL5 and G361 melanoma models wherein PPARGC1A levels were unchanged following chronic BRAF-inhibitor treatment (new **Extended Data Fig. 2b**). In addition, based on our proteomics data, **ii**) EZH2 protein levels become increased in chronic BRAF-inhibitor treated K029A (resistant) cells compared to parental (for **reviewers Figure 1 here**). Furthermore, **iii**) using each of two different EZH2 inhibitors (GSK126 and tazemetostat (EPZ-6438)) restored PPARGC1A level in K029A, A375P and SKMEL3 that had been chronic treated with BRAF-inhibitor (new **Extended Data Fig. 2c and d**). Collectively these data suggests that chronic BRAF-inhibitor treatment increases EZH2 activity (RNA and protein levels) to cause epigenetic silencing of PPARGC1A expression through increasing H3K27 trimethylation marks at the PPARGC1A promoter.

Authors should show similar data with another cell line.

In response to the request, we have now included another high expressing PGC1 α melanoma cell line, SKMEL3. Consistent with the results from K029A and A375P, also SKMEL3 exhibit marked reduction in PPARGC1A expression after chronic adaptation to BRAF-inhibitor (**new Extended Data Fig. 1c**), increased frequency of lung metastasis (**new Extended Data Fig. 1d**), and increased statin sensitivity *in vivo* (**new Fig. 2i**) and *in vitro* (**new Extended Data Fig. 3d**). Thus, we now provide robust data across three melanoma cell lines indicating this as a generalizable effect.

2- To find vulnerabilities within this subset of melanoma cells, authors use a metabolism-focused compound library comparing growth inhibition in drug naïve versus BRAFi-resistant cells. What is the exact rationale behind using this library of compounds?

Given the known role of PGC1 α in controlling mitochondrial biogenesis, its' silencing is expected to reprogram cellular metabolic demands. This prompted us to perform a metabolism-focused small compound library screen to identify molecules capable of specifically inhibiting the growth of these chronic BRAF-inhibitor adapted melanoma cells with reduced PPARGC1A expression. We have revised the text to more clearly state the rationale for the approach.

3- Authors claim increased sensitivity to HMGCR inhibitors in a subset of BRAF inhibitor resistant melanomas throughout but only one compound is tested, Pitavastatin (other than in the initial screen). Key findings should be proven with other drugs to make such statement.

In addition to pitavastatin, we have now also assessed rosuvastatin and lovastatin as additional genuine HMGCR inhibitors. Compared to parental cells, chronic BRAF-inhibitor treatment accompanied with PPARGC1A suppression were consistently also more vulnerable to rosuvastatin and lovastatin (**new Extended Data Fig. 3a,b,c,d**). These new data thus suggest HMGCR-inhibition as a generalizable vulnerability provoked by chronic BRAF-inhibitor treatment and PPARGC1A silencing.

4- Authors show that BRAFi resistant cells that downregulate PGC1a are more sensitive to statins while BRAFi resistant cells that maintain PGC1a levels are not. What makes this subset of melanomas more sensitive to statins? Would silencing of PGC1a induce sensitivity to statins in the other subset of BRAFi resistant cells? In the same direction, do cells with different levels of PGC1a have different sensitivity to statins (naïve to BRAF inhibitor treatment). If statins prevent prenylation of RAB proteins, impairing their function, why are parental cells not sensitive to the effects observed in BRAFi resistant cells in terms of cell adhesion and p-FAK?

We apologize for not making these points clear in the manuscript. Based on our experimental data, the chronic BRAF-inhibitor treated melanomas with silenced PPARGC1A have reduced expression of RAB27A and RAB6B. While their reduced expression is sufficient support unperturbed cell growth, a collateral vulnerability is exposed by targeting HMGCR using statins (pitavastatin, rosuvastatin, and lovastatin). We could rescue the statin effects by downstream supplementation of GGPP (downstream metabolite of the mevalonate pathway), which is an intermediate in the prenylation of RAB proteins and allows their functional membrane targeting. Our data goes further to indicate that combinatorial RAB6B and RAB27A over-expression (but not prenylation mutants) is sufficient to rescue the observed statin sensitivity, and conversely, that targeted suppression of PPARGC1A expression (decreases RAB6B and RAB27A) causes statin sensitivity. While genetic PPARGC1A suppression in naïve A375P and K029A yields statin sensitivity (**Fig. 2d** and **Extended Data Fig 3m**), it is however not observed in naïve SKMEL5 and G361 (**Figure 2 for reviewers**). Hence, the functional consequences of suppressing PPARGC1A expression across these cell lines vary by either of these two

outcomes, which in addition seem to associate with the fate following adaptation to chronic BRAF-inhibitor treatment. We are naturally excited to investigate this fact for mechanistic insights as in BRAFi treatment naïve SKMEL5 and G361 genetic PPARGC1A suppression failed to downregulate RAB6B and RAB27A suggesting other compensatory mechanisms, however, we feel that these mechanistic insights lay outside of the current manuscript.

5- Authors use CCK8 assay to measure cell viability in the drug screen and also to compare sensitivity to statins in BRAFi resistant vs parental cells (Fig 2). Given the data from figure 4 where it is shown that pitavastatin decreases membrane integrin expression and adhesion in BRAFi resistant cells, authors should demonstrate that effects observed using CCK8 assay are due to changes in viability rather than just changes in cell number due to defective adhesion.

The CCK8 assay utilizes tetrazolium salt conversion into an orange formazan dye by cellular dehydrogenase activity, which if dehydrogenase activity specifically is not affected by treatment provides reliable measure of (relative) live cell number, whether adherent or in suspension. Because we did not change culture medium during the CCK8 assay, all cells remained in the wells, and thus these results are expected to include all live cells (adherent and suspended).

6- Data showing a link between MITF, PGC1 α , RAB proteins and sensitivity to pitavastatin is not very conclusive. Authors show that MITF regulates PGC1 α expression and in turn PGC1 α regulates RAB expression, which according to authors drive changes in sensitivity to pitavastatin. Indeed, disruption of either MITF or PGC1 α using CRISPR/Cas9 sensitizes cells to pitavastatin. However, ectopic expression of PGC1 α in MITF depleted cells is not able to rescue the phenotype. Authors suggest that “appropriate expression of each PGC1 α and MITF control HMGR-inhibitor sensitivity” but this explanation is not convincing. Is MITF regulating RAB expression independently from PGC1 α ?

Although MITF contributes to PPARGC1A expression in melanocytes and melanoma cells, additional factors which includes CREB (downstream of cAMP signaling) and muscle MEF2 exemplifies its’ context and multifactorial regulation. While MITF binds DNA directly as a transcription factor, PGC1 α is a transcriptional co-activator that associate with certain transcription factors, including NRFs and ERRs among others relevant to promoting mitochondrial biogenesis. In our view, the simplest explanation is that both MITF and PGC1 α are needed for RAB6B and RAB27A expression, because neither are alone sufficient based on our experimental data and parallels their appreciated roles in controlling pigment cell survival cues and mitochondrial biogenesis, respectively.

7- On the one hand, PGC1 α regulates RAB expression, which is responsible for increased sensitivity to statins in BRAFi resistant cells. On the other hand, authors show that GGPP is able to rescue the effect induced by statins in BRAFi resistant cells, while other metabolites from the mevalonate pathway aren’t, suggesting that impairment of RAB prenylation, and therefore traffic to the membrane, is driving sensitivity to statins. Given that in the subset of melanomas under study PGC1 α levels are very low-same as RAB levels, how is it possible to rescue the effect of statins to the same extend using either RAB6B/27A overexpression or GGPP

supplementation? Also, are RAB6B/27A WT ectopically expressed already prenylated and therefore not sensitive to lack of GGPP due to statins treatment?

We apologize that our initial manuscript did not effectively bring together the mechanistic experiments into a clearly detailed concept on how we believe the collateral vulnerability to statins emerges from PPARGC1A silencing. We have now revised the manuscript to appropriately detail our conceptual findings; and along with point #4 above, we make it clear that silencing of PPARGC1A reduces specific RABs levels, but these cells are capable of growing. However, when subjected to HMGCR inhibition (using statin treatment), these PPARGC1A silenced cells are unable to grow. The growth inhibitory effect of statins in PPARGC1A silenced cells can be rescued by mevalonate or GGPP supplementation, or combinatorial expression of wild-type alleles of RAB6B and RAB27A. Because RAB function requires subsequent prenylation using endogenous GGPP, we surmise that HMGCR inhibition limits the available pool which then is shifted by mass action and used for the overexpressed RAB6B/RAB27A-alleles geranylgeranylation. This is also supported with the use of RAB6B/RAB27A allele mutants that are not prenylated and fail to rescue statin resistance (Fig. 3g, h, 4d and Extended Data Fig 6d, e). We have included now this explanation in the new submitted manuscript and incorporated a model (Fig. 4k).

8- Authors suggest that integrin mRNA expression is modulated by PGC1 α and by RAB6B/27A. They show high mRNA expression of integrins in low PGC1 α expressing cells (both parental cells with sgPGC1 α and BRAFi resistant cells) and high mRNA expression of integrins in BRAFi resistant cells with ectopic expression of RAB6B/27A. How is integrin mRNA expression regulated by PGC1 α ? And by RAB6B/27A? How is this linked to statins sensitivity?

We apologize again that we did not clearly articulate the mechanism by which RAB expression is modulated, and how RABs impact integrin localization and function. Here, we detail experimentally that chronic treatment with BRAF-inhibitor leads to PPARGC1A silencing, which akin to our published work² is linked to increased metastatic behavior. We reported that PGC1 α potentiates transcription of the repressor ID2 that sequesters TCF4, which acts as a transcriptional activator of integrin expression. Thus, reducing PGC1 α co-activation of ID2 results in increased TCF4-dependent transcription of integrin genes. Here we provide experimental evidence that PGC1 α (and MITF) positively controls RAB6B and RAB27A transcription (Extended data Fig. 7), which proteins in turn regulates membrane localization of integrins (Fig. 4f). We have now substantially revised the manuscript to describe the two opposing acting mechanisms of controlling integrin levels, i) PGC1 α acting on ID2 to inhibit TCF4 to reduce integrin levels, and herein reported function, ii) PGC1 α increasing RAB6B and RAB27A levels that allows membrane localization of integrins; and a model is provided as further illustration (Fig. 4k).

Minor comments:

1- Tumours from K029A-R-RAB6B/27AWT and MT grow faster than A-R-LacZ (Fig 3h), how do authors justify these differences?

If compared across the mock treated (vehicle) LacZ, wt RAB6B/RAB27A, and MT RAB6B/RAB27A there were no statistically assured differences in tumor growth based on the standard errors of our measurements. Nonetheless, we have repeated the entire

experiment using comparison also with K029A-R-GFP expressing cells, and this experiment reaffirmed that there are no overt effects on growth by these RAB alleles compared to control, and treatment with statin only affects the growth of control and K029A-R-RAB6B/27A MT cells (Figure 3 for reviewers).

2- Mass spectrometry proteomic data shows a decrease in several RABs but in figure 3 authors only focus on RAB6B and RAB27A. Authors should justify why they focus only in these two.

Our (old Extended Data Fig.1E) indicates that RAB6B and RAB27A are the most significant downregulated RAB proteins; data which is consistent with PPARGC1A melanoma co-expression analyses in TCGA and CCLE (old Fig 3C/D). We have now revised the text to outline this thought process and changed the order of the figures to make this clearer. Old proteomic extended data Fig. 1e is now Fig 3c, and CCLE is Fig 3d, while the analysis of TCGA-SKMC is now in Extended data Fig. 4f.

3- Authors should explain the rationale behind the combinations used in Extended Data Fig. 7a, b)

We have now made it more clear in the text that in addition to RAB6B and RAB27A, also RAB7A, RAB32, and RAB38 were found at lower levels in chronic BRAF-inhibitor treated cells by proteomic analysis, and correlated with PPARGC1A expression levels in melanoma cell lines and tumor biopsies.

4- Key findings should be repeated with another cell line.

(See also Major critique #1 above): SKMEL3 has now been added and shown to replicate the findings with A375P and K029A with regard to silencing PPARGC1A levels and rescue with EZH2i, increased metastatic spread, and collateral sensitivity to HMGCR inhibition.

5- Authors should double check figure citations throughout the manuscript, there are some mistakes. Also, figures should be arranged in the order the panels are cited in the text.

We thank the reviewer for pointing out these errors. We have now corrected the figure citations and aligned the order of the figures with the flow of the text.

6- Authors should indicate the N of each experiment.

We now have now detailed replicate “n” for each experiment.

7- Authors should make sure the bibliography is adequate. For instance, in this sentence the references cited are not relevant “While there are preclinical studies suggesting that HMGCR inhibitors alone or in combination with other drugs effectively inhibit melanoma cell growth 22,27”.

We now have made sure that appropriate and correct references are given throughout the manuscript text.

Reviewer #2:

In this manuscript, Liang et al identify a subset of BRAF inhibitor-resistant melanomas that emerge during chronic drug treatment and show that they are associated with decreased levels PPARGC1A, because of increased levels of H3K27me3 and reduced levels of H3K27ac marks across the PPARGC1A promoter region. PPARGC1A silencing in these tumors are associated with metastatic traits but leads to sensitivity to HMGCR-inhibitor treatments. They also show that suppression of PPARGC1A results in lower RAB6B and RAB27A levels that in response to HMGCR-inhibitor treatments reduces cell surface trafficking of integrin and downstream FAK signaling (which are elevated in BRAF inhibitor-resistant cells). The topic of drug resistance in melanoma

and the challenge to overcome such resistance is clinically significant. However, there are major limitations that diminishes the enthusiasm for this work.

We thank the reviewer for the summary and thoughtful itemized critique below. We have addressed the concerns that limited the enthusiasm in the following itemized responses.

1 - In the introduction of the manuscript, the authors only review the impact of MAPK/ERK pathway PI3K/AKT pathway as known mechanisms of BRAF inhibitor resistance in melanoma. Then, they motivate their studies, based on the idea that "It is currently not entirely known whether targeted BRAF-drug resistance is widespread, or only found pre-existing within a small population of cancer cells. Importantly, it is not understood whether there are any underlying mechanisms that can be used to inform treatment resistance, and whether co-therapeutic strategies can prevent or target emerging resistance." However, there are many impactful papers published in the past 5 years that have investigated the role of epigenetic adaptation and differentiation state plasticity on MAPK inhibitor resistance. The connection of PPARGC1A and the differentiated (MITF^{High}) state has also been previously discussed. The authors should improve the introduction to the previous studies and specifically review the more recent literature of the BRAF/MEK inhibitor resistance in melanoma that are more relevant to their work.

We apologize for not providing more nuanced and inclusive references to previous work in this rapidly moving field. We have naturally amended with additional citations that are specific to what is known about acquired resistance and are relevant clinically. We have also added references to reviews detailing adaptive/epigenetic as well as metabolic adaptation, concepts that we are aligned with but have not yet proven to be clinically relevant.

2- Most studies in this manuscript are performed in one or two cell lines. This raises questions and concerns about the generalizability of the findings. This is especially important, given that diverse cell lines do behave differently with respect to induced changes in the expression of differentiation markers (e.g., MITF) and PPARGC1A following drug resistance. Some cell lines show increased expression and in only some cell lines (that are selected for this study), PPARGC1A levels drop following MAPK inhibitor treatment.

We entirely agree that the burden is on us to demonstrate the generalizability of our findings, and also brought up as a concern by Reviewer #1. We had essentially used two cell lines, A375P and K029A for the bulk of the work. We have now added a third cell line, SKMEL3, which results essentially parallels that of the prior two. In addition, we contrast the findings with two alternate cell lines SKMEL5 and G361 wherein BRAF inhibitor resistance does not associate with PPARGC1A suppression, and a collateral HMGCR inhibitor sensitivity.

3- The authors focus their study of chronic BRAF inhibitor resistance on PPARGC1A, based on the finding that the PPARGC1A levels drop significantly as shown in Fig. 1c, 1d. However, the drop of PPARGC1A levels happen much earlier than the emergence of proliferating drug-resistant cells. So, does the PPARGC1A depletion lead to drug tolerance at earlier timepoints? If so, shouldn't we expect synergy between pitavastatin and BRAF/MEK inhibitors at least in cell culture?

The graph in Fig 1c/d details PPARGC1A mRNA levels and normalized to treatment-start cell numbers. The fact that the drop in PPARGC1A precedes rapid expansion of cell numbers is because the emerging cells adapt to BRAF-inhibitor treatment while the majority are unable to grow and dies (slope goes below 1). Towards possible synergy between HMGCRi (pitavastatin) and BRAFi, combination treatment showed no more inhibitory effect than the additive of statin and BRAFi alone during 5 days of treatment, indicating no synergy

Fig 4. The effect of pitavastatin and BRAFi in the cell survival of cells. Ten thousand K029A cells were seeded into 96-well plate and treated with pitavastatin and/or BRAFi at indicated concentration for 5 days. Cell viability was measure via CCK8 kit. The line of additive is a mathematical addition of the inhibitory effect of statin and BRAFi respectively (n = 6).

Fig 5. The effect of pitavastatin and BRAFi in the cell survival of cells. A thousand K029A cells were seeded into 6-well plate and treated with pitavastatin and/or BRAFi at indicated concentration for 3 weeks. Cells were fixed with methanol and stained with crystal violet.

exciting question indeed. We have performed western blot analyses to determine levels of H3K37me3. We cannot detect any global changes within A375P and K029A as a function of chronic adaptation to BRAF inhibitor (For reviewers **Fig. 6**), and thus that general metabolic differences such as one carbon metabolism and availability of methyl group donors. Because we see changes in EZH2 levels, and EZH2i is able to reverse the PPARGC1A silencing, it is likely only a subset of promoters are affected. However, exhaustive ChIPseq analyses to detail which genes are affected will have to be outside of the current manuscript because of scope.

effect between statin and BRAFi (**for reviewers Fig. 4**). However, and consistent with that PPARGC1A silencing drives a collateral vulnerability to HMGCR-inhibition, co-treatment prevented emergence of resistant cell colonies (chronic adaptation; no A BRAFi resistant cells formed with the combination treatment of 1 μ M pitavastatin and 1 μ M BRAFi together for three weeks)(**for reviewers Fig. 5**).

4- The authors conclude that “epigenetic” silencing of PPARGC1A drives metastatic traits and leads to collateral sensitivity to HMGCR-inhibitor treatment. This is based on the measurements of two histone marks H3K27me3 and K3K27ac and their changes following chronic drug resistance (Fig. 1g). However, as mentioned above, PPARGC1A silencing begins as early as a few days post-drug treatment in cells (Fig. 1c). So, do these histone marks change within the same time scale (i.e., after a few days)?

The slope of the PPARGC1A decrease levels out at 1-2 weeks in K029A and more than 3 weeks in A375P cells. Whether this dynamic is reflected also in H3K27-marks is a very interesting question indeed. However, for technical reasons relating to the very limited amount of material available at earlier time points, we were not able to perform ChIP analysis across the PPARGC1A promoter region. Specifically, after one week of BRAF-inhibitor treatment there is a very low cell number and insufficient for this assay in our hands.

5- What makes such epigenetic changes possible in some cell lines but not in others? Are the epigenetic modifications that result from H3K27me3 and K3K27ac changes affect only PPARGC1A, or a larger pattern of gene expression associated with the drug-resistance state?

This is a very interesting and

Fig 6. Western blots of H3K27Me3, H3 and Tubulin in the parental and BRAF-inhibitor chronic adapted K029 and A375P cells.

6- *In vivo* studies presented in Fig. 2g-2k are done in various lengths from 9 days to >20 days. Since the authors compare these data to make conclusions about the underlying conditions for pitavastatin efficacy, it is surprising to see that they have not performed these experiments for the same durations. It is clear, for example, that 10 days of treatment with pitavastatin in Fig. 2h does not provide significant efficacy in A375P-R tumors and treatments need to continue for >20 days for the drug to be effective. So, why did the authors terminate the experiment on A375P tumors after only 9 days (Fig. 2i)?

Yes – we have to use different timelines for various melanoma cell lines because of their variant growth rates, and the BRAFi-R cell lines are different as well. According to our approved DFCI IACUC animal protocols, we have to end mouse tumor implantation experiments when the largest tumor exceeds 1000mm³. Thus, the BRAF-inhibitor resistant tumors in Fig 2h grew slow and took around 20 days after treatment to reach similar tumor volumes as observed in Extended Fig .3o. The data in Extended Fig .3o clearly indicates that statin treatment has no anti-tumor effect in parental A375P cells.

Reviewer #3:

The role of PGC1a in tumorigenesis has been addressed in many previous publications, including in several studies on melanoma. In particular, Puigserver and his group have shown before that PGC1a is essential for survival of a subset of melanoma cells and that it regulates integrin expression and invasiveness / metastasis formation. In the present manuscript, Puigserver and colleagues extend these findings and reveal that, in a subset of melanoma cells emerging upon chronic BRAFi treatment, PGC1a expression gets suppressed, which is associated with more aggressive (invasive and metastatic) features in human cell lines. Using a metabolism-focused compound library screen, the authors identify BRAF-inhibitor resistant melanoma cells with silenced PPARGC1A to be sensitive to HMGCR-inhibitors in vitro and in vivo. Mechanistically, the authors show that PGC1a regulates RAB6B and RAB27A, which are important for sensitizing PPARGC1A low BRAFi-resistant cells to HMGCR-inhibitors (such as pitavastatin) also in vivo. Further, pitavastatin reduced membrane association of RAB6B and RAB27A affecting cell surface integrin receptor association and downstream FAK activation, leading to reduced cell growth. This is an interesting paper that convincingly links PGC1a to targeted therapy resistance in a subset of melanoma cells. However, some important issues need to be addressed before publication.

We thank the reviewer for the scholar summary of our work, and the critique that we have addressed below.

Major points:

- *Two out of four BRAFV600E-mutant melanoma cell lines show reduced PPARGC1A mRNA expression in BRAFi resistant cells (Ext.Fig.1b). However, the authors do not show whether BRAFi-resistant cell lines with silenced PPARGC1A (A, A375P) are indeed more aggressive than BRAFi-resistant cells displaying increased PPARGC1A (G361, SKMEL5), as predicted from the analysis of patient data (Fig. 1b). The authors should extend the data shown in Fig. 1e by investigating these additional cell lines in vivo.*

We have now conducted the experiments across three (3) cell lines yielding reduced PPARGC1A mRNA expression following chronic BRAF-inhibitor treatment; K029A, A375P and newly added SKMEL3 (Fig. 1c, d; Ext.Fig.1c). As BRAF-inhibitor resistant cells, all three cell lines exhibited more metastasis-like behavior in vitro and in vivo (Fig. 1e; Ext.Fig.1d, e). In contrast, the invasive characteristics of BRAF-inhibitor resistant cells generated from G361 and SKMEL5 cells were not altered compared to their parental (treatment naïve) cells (Extended Fig. 1e).

• *On the same token, the authors conclude from the above-mentioned experiments, that their main finding (“Epigenetic suppression of PGC1a (PPARGC1A) causes collateral sensitivity to HMGCR inhibitors within BRAF-treatment resistant melanomas” (Title)) only applies to a subset of melanoma cells. To substantiate this important statement, the authors should use at least three cell lines representing this subset of BRAFi-resistant melanomas for their key experiments. Many experiments have been done with just one cell line, some with two.*

As indicated also in the previous point, we now demonstrate across three (3) independent melanoma cell lines, K029A, A375P and SKMEL3 cells that PPARGC1A was downregulated upon chronic BRAF-inhibitor treatment in a manner that could be reversed by EZH2 inhibitor, acquisition of metastasis-like behavior in vitro and in vivo, and HMGCR-inhibitor sensitivity.

Furthermore, the authors should show that in cell lines not representing this specific BRAF-resistant subset (such as G361, SKMEL5), PPARGC1A is not epigenetically suppressed by EZH2 and that the proposed downstream effectors pFAK, RAB6B and RAB27A are differentially expressed in K029A, A375P, etc., as opposed to G361, SKMEL5.

In this revised manuscript version, we show that expression levels of PPARGC1A in G361 and SKMEL5 are among the highest across melanoma cell lines¹. Furthermore, EZH2 levels changed in K029A, A375P and SKMEL3 upon chronic treatment with BRAF-inhibitor, while this was not observed in BRAF-inhibitor resistant G361 and SKMEL5 cells (**Extended Fig. 2b**). Likewise, RAB6B and RAB27A levels were not changed in resistant G361 and SKMEL5 cells (**Extended Fig. 5a**).

• *Low PPARGC1A expression has been associated with emergence of resistant cells, but decrease in PPARGC1A expression occurs way before resistant cells emerge (Fig. 1c,d). This rather speaks against a direct connection between these two processes. How do the authors explain this? Is, for instance, the regulation of RAB6B and RAB27A downstream of PGC1a indirect and delayed compared to the downregulation of PPARGC1A upon BRAFi treatment?*

We fully agree that the emergence of BRAF-inhibitor resistance is only associated with PPARGC1A downregulation, and not implicit because G361 and SKMEL5 maintain high levels when resistant (also raised by Reviewer #2 (point 3)). We never suggest that PPARGC1A downregulation causes BRAF-inhibitor resistance, but rather we demonstrate that PPARGC1A downregulation causes a collateral vulnerability to HMGCR-inhibitors that can be exploited by statins. We have revised the text to detail this fact more clearly.

• *The authors propose a link between epigenetic regulation of PPARGC1A and its role in resistance formation. Does the kinetics of H3K27me3 occupancy at the PPARGC1A promoter follow its expression during BRAFi treatment? DZNep is not specific for EZH2, the use of a more specific inhibitor would be preferable. Moreover, would HDACi treatment interfere with resistance formation? Such experiments are relevant because the authors emphasize the role of epigenetic suppression of PGC1a in the processes described here.*

Assaying the dynamics of H3K27me3 changes is technically challenging because of the rather high number of cells needed to perform ChIP analysis, and at early time-point there is not many cells available (similar concern raised by Reviewer #2 (point 4)).

Towards the generalizability of EZH2 involvement, we have included another clinically used EZH2 inhibitor, tazemetostat that exhibits similar results compared to DZNep (**Extended Fig. 2d**). Thus together with the observed changes in EZH2 expression, this suggests a potential mechanism for an involvement of EZH2 function in PPARGC1a downregulation during BRAF-inhibitor treatment.

Since inhibition of EZH2 mediated trimethylation could potentially restore PPARGC1A expression in the BRAF-inhibitor adapted cells, we assume the downregulation of PPARGC1A was mainly dependent on the regulation of H3K27me3. However, regarding the role of HDACs, and use of HDAC inhibitors specifically, it is a very different and potentially complicated extension of this work. Histone deacetylases are likely involved, but whether it is dependent on HDAC1 or SIRT6, or some other HDAC remains unknown and specific inhibitors for mechanistic studies are not available.

- *To strengthen the functional link between PGC1 α and RAB27A and RAB6B, rescue of pitavastatin-treated cells by overexpression of RAB27A and RAB6B (Fig. 3, Extended Data Fig. 6) should also be done on a sg-PPARGC1A background.*

This is an important experiment which we now have performed. Over-expression of RAB6B and RAB27A in sgRNA PPARGC1A K029A cells indeed rescued pitavastatin sensitivity (new Extended data Fig. 5d, e).

- *Related to Figure 4, the changes in pFAK levels should be shown in vivo (in settings as presented, e.g., in Fig. 3h). Furthermore, Marin-Bejar et al (Cancer Cell 2021) have previously reported a functional association between FAK expression and resistance formation to targeted therapy. This should be discussed.*

Now we analyzed pFAK and total FAK levels of K029A-R tumors (from mice injected with K029A chronically adapted cells). Pitavastatin treatment significantly decreased pFAK levels in tumors (new Extended data Fig. 10e).

- *The relationship between PGC1 α and MITF (Ext. Data Fig. 5) could be strengthened and should be further discussed in the context of existing literature. It is intriguing that MITF and PPARGC1A, respectively, cannot rescue each other in the experiments presented in Extended Data Fig. 5d,e. How do the authors explain this, given the close relationship between MITF and PPARGC1A? The statement on p.5 “that appropriate expression of each PGC1 α and MITF controls HMGCR-inhibitor sensitivity, ... presumably by regulating RAB6B and RAB27A levels” could be experimentally addressed.*

MITF binds DNA directly as a transcription factor, while PGC1 α is a transcriptional co-activator that associates with certain transcription factors (i.e. NRFs and ERRs). Because neither are alone sufficient to rescue RAB6B and RAB27A expression, appropriate expression of both, and potentially others unidentified factors are required. To avoid over-interpretation, we have revised this statement on page 5; similar also concern raised by Reviewer #1 – point 6.

- *In general, the methods are poorly described. For instance, the authors should explain in the method section how they analyzed the publicly available datasets shown in Fig. 1a, b. Further, they should specify how long they maintained the specific cell lines under BRAFi treatment, and whether the phenotype remained stable over time – especially, when the cells were modified.*

In this revised version of the manuscript, we have provided more experimental details that should allow sufficient information to reproduce our data.

- *Statistics are insufficiently described. Almost all figures are displayed as bar graphs without ever mentioning the number of replicates shown and without displaying the actual data points. Furthermore, there is no information on how many times representative western blots have been repeated. This must be specified.*

This information is now appropriately specified in the revised manuscript.

- The data presented in Fig. 4d need to be quantified.

We have quantified the data (for reviewers Fig. 7). Pitavastatin treatment significantly increased the soluble part of RAB27A and RAB 6B, indicating the de-prenylation of those two proteins. Since the volumes of lysates of total, membrane, soluble parts were different due to the procedure of isolation, the protein levels are incomparable between different parts.

Minor points:

- In Fig. 1f, the authors show that overexpression of PPARGC1A abolishes cell migration in BRAFi-resistant K209A cells, indicating that the migratory effect is dependent on PGC1a. The authors should show validation of the overexpression in this experiment.

We have now included the level of re-expressed PPARGC1A in Extended Fig. 1f.

Fig 7. Quantification of western blots in Fig. 4d.

- In the figure legend to Fig. 1g and Fig. 3j, the authors named the cell line K029S. Is this a different cell line than K029A?

K029S was indeed a typo, should read K029A and that has been changed.

- The figure order is confusing, making the paper (which is anyway written in a very compact manner) difficult to read at certain places. Authors should rearrange figures (especially Extended Data Figures 3 and 4). Further, they should cite the figures in an alphabetical order in the text and mention all the figures in the text (they did not refer to Fig. 2b in the text, for instance).

We now have all the figures in order with flow of the text.

- In Fig. 3j, the authors show that silencing of RAB27A and RAB6B individually sensitizes K029S (A?) cells to pitavastatin. The authors claim that double silencing further increases statin sensitivity, however, this is not obvious from the graphs presented.

We agree fully with the reviewer, and we have changed the text accordingly.

- On p.5, “we searched our mass spectrometric-based proteomic data which evidenced a decreased amount of RAB6B and RAB27A in chronic treated BRAF-inhibitor resistant A cells compared to their parental counterparts” should refer to Extended Data Fig. 1e, not Extended Data Fig. 1d.

This has now been corrected, and we apologize for these errors in putting together the initial manuscript submission.

Reviewer #4:

The authors determine that epigenetic suppression of PGC1alpha is associated with an aggressive subset of BRAF-inhibitor resistant melanoma. They go on to identify that tumor cell lines with repressed PGC1alpha expression are particularly sensitive to statins. Mechanistically they invoke that statin exposure leads to depletion of GGPP pools, which results in the decrease in the isoprenylation of RAB proteins. In particular, they argue that RAB6B and RAB27A play a key role in the growth and survival of this subset of melanoma. They

further suggest that there is a decrease in integrin complexes and reduced FAK activity. The authors have made a substantial contribution; however the direction of research is not always rationalized and data interpretation is not always considered in full. Issues associated with attention to detail are also problematic.

We thank the reviewer for the concise overall summary and well-pointed critique that we have addressed (see below).

1. The entire manuscript is performed on two parental melanoma cell lines (K029A, A375P) and their BRAF-inhibitor resistant counterparts that demonstrate a 2- to 4-fold decrease in mRNA expression of PPARGC1A, a transcriptional coactivator and master regulator of mitochondrial biogenesis. The results would be significantly strengthened with additional data in relevant melanoma patient-derived model systems, such as organoids or PDXs. Evaluation in immune-competent melanoma models would also add weight. Results in these types of relevant models would also support the authors' conclusion that the preclinical data from this manuscript supports a clinical trial. Without these additional models, the results are quite limited as they largely focus on K029A cells.

This is a critique also raised by the previous reviewers. We have now included additional experiments in A375P and SKMEL3 (new added human melanoma cell line) that parallels the previous findings in K029A cells, and hence, key data is replicated across three human melanoma cell lines. In addition, we extend the analysis to additional melanoma cell lines on chronic BRAF-inhibitor treatment resulting in PPARGC1A silencing and arising collateral vulnerability to HMGCR inhibitor treatment (**Extended data Fig. 3a-i and Fig. 2g-k**). The use of PDXs for these experiments is hampered by the fact that chronic treatment would have to be performed in vitro, and hence, as such would no longer be considered a PDX, but rather a short-term melanoma culture; data that would not be expected to be different than established cell lines. Finally, congenic murine melanoma cell lines generated by BRAF(V600E) don't express appreciable levels of PPARGC1a, and those that (i.e. B16) don't have BRAF(V600E).

2. How does reduced PPARGC1A expression lead to enrichment in plasma membrane, ECM and integrin proteins, as well as a decrease in RAB protein expression, as shown in Ext Fig 1e? Is this at the level of transcription? RNAseq and ChIP-seq data would provide mechanistic insight.

Based on this concern, and Reviewer #1's point 8, we have substantially revised the description on how we conceptualize the relationship between PPARGC1A suppression, decreased RAB function, and membrane integrin enrichment. We previously published that reducing PGC1 α reduces ID2 ability to inhibit TCF4-dependent transcription of integrin genes (Luo and Lim et al. Nature 2016). Here we provide experimental evidence that PGC1 α (together with MITF) positively controls RAB6B and RAB27A transcription (**Extended Data Fig. 7**), which in turn regulates integrin membrane localization (**Fig. 4f**). Reducing PGC1 α levels negatively affect RAB protein levels, and thus integrin membrane localization; however, there are higher total integrin levels. We also hope that our new model helps illustrate this conceptually (**Fig. 4k**).

3. Why is the K029A control not shown (Fig 2g)? Experiments conducted at the same time should be displayed on the same graph (e.g. Fig 2h and 2i; Fig 2j and 2k).

Parental K029A cells are less able to nucleate tumors and grow in vivo compared to their BRAF-inhibitor resistant counterparts. There were no tumors formed when 1 million parental cells were implanted s.c., but the BRAF-inhibitor resistant cells readily formed tumors at this implanted amount. In order to have xenograft

tumors, we injected 10 million parental K029A , and these tumors were not sensitive to statin treatment (shown in **Fig. 2k**).

4. Ext Fig 4: The authors have not provided data to show that the metabolites were taken up by the cell. Thus, either this data needs to be shown or this caveat must be included in the text as part of data interpretation.

The reviewer raises an important point. It has, however, been shown that farnesol^{3,4}, cholesterol⁵, CoQ10⁶ are taken up by cultured cells. Neither of these supplemented biomolecules could rescue the functional effects from HMGCR-inhibition, but addition of GGPP could. We have now included the references above in the text to avoid any second thoughts about the validity of our approach.

5. Fig 3c and 3d: The authors state that melanoma biopsies (TCGA: SKCM) and cell lines (Broad: CCLE21Q1) datasets show RAB6B and RAB27A mRNA expression was highly correlated with PPARGC1A mRNA levels. However, other RABS (RAB38, RAB7A, RAB32) are also highly correlated, yet they are dismissed. Other RABS are also downregulated in BRAF inhibitor resistant cells compared to parental (Ext Fig 1e). The authors need to show the functional role of these RABS. This is important because neither 6B or 27A was functionally important on its own, suggesting a class effect. It is likely that the loss of isoprenylation of several RAB proteins is contributing to pitavastatin activity. The rationale for focusing only on two RAB proteins is not justified. If it is a class effect of all isoprenylated proteins, including several RABs, then that is the message that needs to be delivered. Emphasizing the two RAB proteins alone as holding key functionality is not supported by the data as the authors have not shown that the other RAB proteins don't also contribute to the effect of pitavastatin exposure.

The reviewer recognizes the strong correlation between PPARGC1A levels with that of RAB6B and RAB27A across each of the TCGA and CCLE melanoma datasets. However, the additional RAB proteins RAB38, RAB7A, and RAB32 also did show a correlation. Our pursuit of RAB6B and RAB27A was also informed by our proteomic data, where these two proteins were the highest correlated, and to which end, we have now rewritten that section of the manuscript, and reordered the figures as well; this concern was also raised by Reviewer #1 - minor point 2/3. It is conceivable that other RABs may play a functional role within PPARGC1A suppressed cells, however RAB27A and RAB6B are sufficient to rescue statin sensitivity.

6. Are 6B and 27A preferentially affected at the level of isoprenylation in response to pitavastatin? Evaluating levels of prenylated and unprenylated forms of several RAB protein levels in the presence or absence of pitavastatin treatment would determine if there was preferential loss in the isoprenylation of specific RABs, which may explain why these RABS (6B, 27A) are functionally critical. Mass spec or other types of isoprenylation assays may be useful here.

We show in **Fig. 4d** that statin treatment changes RABs from prenylated form to unprenylated form (detach from membrane) and in **Fig. 3g** overexpression of wild type RAB6B and RAB27A alleles but not their prenylation site mutants could rescue the observed statin-sensitivity with BRAF-inhibitor resistant cells. In addition, RAB6B prenylation cause a molecular shift in SDS-PAGE/western blot analysis (**Fig. 4d, Extended Data Fig. 10a**). This data together indicates the prenylation of RAB6B and RAB27A plays a functional role in statin-induced BRAF-inhibitor resistant cell death. We agree that mass spectrometry would strength the biochemical data, but the gel mobility shift and the use of prenylation deficient RABs mutants, in our view, support the role of prenylation in response to HMGCR-inhibition by statins.

7. Information regarding RAB6B and RAB27A need to be provided so the reader might understand better why these two RABs are particularly functionally important for the growth and survival of these cells. It is hard to understand why ectopic RAB6B is so important in combination with RAB27A in terms of the rescue effect. This is particularly difficult to understand given that RAB6B appears difficult to ectopically express in the presence of RAB27A (e.g. Ext Fig 6d), or visa versa (Ext Fig 8c). Was RAB6B ectopic expression more robust at the protein level on its own and/or when combined with other RABS (e.g. 7A, 32, 38; Ext Fig 7 only shows mRNA). As written the manuscript is difficult to understand because the authors don't provide rationale for why they are getting the results they see with one RAB vs another. Perhaps providing more background would be instructive. Are these RABS essential to all melanoma cell growth and survival or for only this subset of melanoma? Etc. Dialogue regarding the results would be helpful in terms of rationalizing data interpretation.

We apologize for not including sufficient information regarding the experimental rationale and RAB6B and RAB27A background (a concern shared by Reviewer #1). Our proteomic data clearly suggests that RAB6B and RAB27A are two RAB proteins most affected within BRAF-inhibitor resistant cells, and since neither of these alone are sufficient to rescue the observed collateral vulnerability to HMGR-inhibition, but their dual re-expression is, it seems like the most rational explanation that they cause the sensitivity. Furthermore, we provide data indicating that each of PGC1a and MITF are involved in their regulation on the mRNA level, and that these two transcriptional regulators are suppressed on the BRAF resistant cells. Finally, we have new data indicating that the observed statin sensitivity from sgPPARGC1A editing can be rescued by RAB6B and RAB27A re-expression as well.

8. To help resolve and provide clarity, the authors need to evaluate whether GGTase inhibitors phenocopy pitavastatin. Are BRAFi-R cells also sensitive to geranylgeranyl transferase inhibitors? We agree with the reviewer that this is an important point.

There are two Rab geranylgeranyl transferases (GGTases): GGTase-I and GGTase-II. Only inhibitors to GGTase-I are currently available; none are commercially

available against GGTase-II. GGTase-I inhibitors show similar effects on parental and BRAFi resistant cells (**Fig. 8 for reviewer**). The only reasonable explanation is that GGPP rescues the effect of HMGR-inhibitors statin through the activity of GGTase-II.

9. Fig 3e, 3f, Ext Fig 5a, Ext Fig 5b: i) What happens to the mRNA expression of the other RAB proteins under these conditions (i.e. CRISPR PGC1alpha, CRISPR MITF, overexpression of PGC1alpha)? Are other RABs similarly regulated as RAB27A and RAB6B? ii) What is the relative growth rate of these cells? Could the effects on expression of the RABs be a reflection of relative cell proliferation/cell death under these various conditions? For example, is the change in RAB expression due to changes in PGC1alpha expression (cause) or due to an indirect effect on cell proliferation (consequence). Knowing the cell proliferation/cell death of these populations will help to determine whether the effect is a cause or consequence.

i) We have assessed the effects of sgMITF on RAB7A and RAB32 levels using qPCR (for the Reviewers Fig. 9), and it seems that these are affected similarly to what we found for RAB6B and RAB27A.

ii) Towards effects on growth rates, we show below the variant effects of chronic BRAF-inhibitor treatment on the resulting growth of K029A and A375P. While K029A-R cells grow faster compared to the parental (untreated) cells, A375P-R have no appreciated difference in growth compared to the parental cells

(Fig. 10 to the reviewers). However, we observed similar effect of PGC-1 α on the expression of RABs and statin sensitivity (Fig. 3j and Extended Fig. 7a) across these two cell lines.

Fig. 9: Expression of RAB7A and RAB32 in K029A parental cells with MITF knockout by qPCR (n = 3).

Fig. 10: Growth curves of K029A and A375P cells (parental and BRAFi chronically adapted). 0.1 million of indicated cells were seeded into p-100 dishes and cultured with growth medium. Cell numbers were counted 24, 48 and 72 h after seeding (n = 3).

10. Fig 3g: Does ectopic expression of additional RAB proteins further the rescue effect seen with ectopic expression of RAB6B+RAB27A? Only RAB6B and the other RAB proteins was interrogated.

Compared to parental cells, we found that re-expression of RAB6B+RAB27A in BRAF-inhibitor resistant cells affords full rescue against HMGR-inhibition using statins (old Fig. 3g now fig, 3f and Extended Fig. 6e). It is conceivable that additional RAB proteins may modify statin sensitivity, however RAB6B and RAB27A are consistently downregulated and these afford rescue. We think that a detailed stoichiometry analysis of all the different RABs goes beyond this manuscript and will be investigated in future studies, especially if specific RAB inhibitors would become available.

11. Fig 4a: The authors indicate HMGR as up-regulated within the figure, but don't explain this in the text as an expected change due to the well-established feedback response to statin exposure that leads to the induction of several mevalonate pathway genes, including HMGR. It is surprising that other genes in this metabolic pathway don't appear to be up-regulated, yet they (like HMGR) should serve as positive controls. Loss of this feedback loop is associated with increased sensitivity to statin exposure (for review see PMID 32887721). If these mevalonate pathway genes are not up-regulated, then this loss of feedback may also be contributing to pitavastatin sensitivity. These mevalonate pathway genes need to be highlighted in the volcano plots as positive controls, e.g. HMGRS1, LDHA, INSIG, etc.

We thank the reviewer for bringing up this concern and agree that the notion of cholesterol-dependent feedback might contribute to statin sensitivity. However, exogenous addition of cholesterol did not rescue, nor worsen, the observed HMGCR-inhibitor sensitivity (**Extended data Fig 4b**); suggesting that during the conditions of our assays it may only have a limited role, if any.

Furthermore, in the proteomic data other enzymes within the mevalonate pathway, such as FPPS, HMGCS1, PMVK, MVK, were in contrast to HMGCR not more than marginally changed (**Fig. 11 for the Reviewers**). It is well known that SREBPs transcriptionally regulate HMGCR levels, and in the presence of HMGCR-inhibitor, they become activated due to lower cholesterol levels in the cell. In

addition, post-transcriptional regulation of HMGCR through ubiquitination is also important when metabolite fluxes occur in the mevalonate pathway⁷.

Fig 11. The proteins in mevalonate pathway highlighted. The left figures are from Fig.4 a, with all the proteins illustrated and the right figures with only proteins in mevalonate pathway illustrated.

12. Fig 4a, 4b: Are the changes in BRAFi-R cells due to increased genomic instability compared to parental cells? Would similar changes be seen in response to any perturbagen? Is the induction of DNA damage required for pitavastatin-induced cell death? Statins do not directly damage DNA. The authors need to frame the results into context and provide the reader guidance on how to interpret these data, rather than just present the results.

Because we can reverse the silenced PPARGC1A levels by EZH2-inhibitor treatment, and sgPPARGC1A is sufficient to render the cells sensitive to HMGCR inhibitors, we don't believe that genomic instability has any major effect on what we report here.

13. Fig 4c: As the authors see evidence of Parp cleavage (Fig Ext 8C) it is important to determine what fraction of cells are undergoing cell death by gating on sub-G1 in these flow cytometry experiments. Also, it is the industry standard to show 3 independent measures of apoptosis, e.g. TUNEL, Annexin V staining, TMRE as a measure of mitochondrial potential, etc.

We have assessed different apoptosis, necrosis and ferroptosis inhibitors, however, none of them could rescue the portion of statin-induced cell death that we observe. It is formally possible that the cell death that we measure is a result of inability to grow, and imbalances that leave the cells to undergo non-programmed cell death. We merely use PARP cleavage as an indicator for cell death. We have changed apoptosis for cell death in the text accordingly.

14. Fig 4e. *“This approach revealed that pitavastatin treatment reduced cell surface localization of a large number of integrins within the chronic treated BRAF-inhibitor resistant cells, which were correspondingly retained by RAB6B and RAB27A over-expression (Fig. 4e, Extended Data Fig. 10b).”* However, the data show that pitavastatin reduced about half of all integrins (Fig 4e). Indeed, the two used for further validation are decreased (A5) and increased (B1) in this figure. The scale is shallow, which could be contributing to the display being misleading? This needs to be addressed either in the text and/or the data display.

We have now addressed this in the revised manuscript. Although integrin-b1 re-distribution was significantly changed by Western blot assays, the change within the proteomic data was relatively moderate. To be consistent with across both assays, we now removed integrin-b1 from **Fig. 4f** and **Extended Data Fig. 10c** to make the information clearer and detailed these changes in the text.

15. *The introduction of NAC into the experimental rationale is missing. It comes out of the blue. Isn't NAC really serving as a positive control rather than a discovery aspect of the work? Rationale is needed for why suddenly an antioxidant is being evaluated. It is important to remember the broad readership of Nature Communications.*

The reason was to introduce a detachment-dependent cell survival pathway described by Joan Brugge's group⁸, which has been investigated with regard to melanoma metastatic spread by Sean Morrison's laboratory⁹. Because we attribute increased metastatic propensity within our BRAF-inhibitor resistant cells with silenced PPARGC1A, the concept was to determine whether NAC treatment would afford further increase survival benefit during detachment. Interestingly, the BRAF-inhibitor resistant cells had elevated survival compared to their parental counterparts that was not improved by NAC treatment, while the parental cells exhibited more survival (**Figure 4i/j**). We think this data provides a suitable and exceedingly interesting parallel to the work of Sean Morrison's laboratory that may spur additional work by others as well.

16. *Also, why is cell survival under de-attachment growth conditions (anoikis) being measured? Rationale again not provided. What about measuring classic apoptosis under regular growth conditions on adherent plates? The latter half of Fig 4 seems distant to the main thrust (see title) of the manuscript. Is this tangent necessary? If so, rationale is required. Even the authors state, “These data indicates that resistance to anoikis driven by elevated integrin receptor expression and FAK signaling is functionally separable from the collateral sensitivity to HMGCR inhibitors seen in chronic treated BRAF-inhibitor resistant melanoma cells with silenced PGC1 expression.” Perhaps a working model figure would help to add clarity.*

Coupled to our response to #16 above, and based on work from Joan Brugge's laboratory and others^{10,11} clearly have shown that integrin signals, and downstream FAK signalling, provides growth survival cues that are able to overcome detachment induced apoptosis (anoikis). We have now also added a model to conceptualize our results.

17. *Abstract: “PGC1alpha suppression drives transcriptional downregulation of RAB6B and RAB27A levels...”* Changes in mRNA expression does not necessarily mean that PGC1alpha is regulating these genes at the level of gene transcription. If so, the authors need to show how. What is the mechanism?

While we do see changes in RAB6B and RAB27 RNA levels following modulation of PGC1a, we don't formally know if the effects are direct and transcriptional, although very likely. We have therefore changed the sentence to now read “Mechanistically, PGC1alpha suppression causes downregulation of RAB6B and RAB27A mRNA levels, whereby ...”

18. The experimental datasets were not provided as supplementary tables, e.g. Fig 4a, 4e, Ext Fig 1e, Ext Fig 2a, Ext Fig 10b. This is important as this is the primary data from which further validation is shown throughout the manuscript.

The experimental data is now provided in supplemental Excel data spreadsheets

Minor:

1. As ITGB5 is further studied, it would be appropriate to highlight and label the data point corresponding to this protein in Ext Fig 1e showing MS proteome data of BRAFi-R vs parental cells.

We believe the reviewer indicated ITGA5 instead of ITGB5, and we now have ITGA5 highlighted (new Fig. 3c).

2. It is important to mention how the drug treatment of pitavastatin used in mouse experiments (1mg/kg; see Fig 2g-k) relates to the dosing of pitavastatin used to lower serum cholesterol in humans. It is also important to relate this dose to that used for in vitro studies, in terms of micromolar equivalents. This provides the reader rationale as to why the doses used in this manuscript were chosen and how they relate to those used clinically and to levels achievable in human plasma.

We agree with the reviewer that a discussion regarding the statin doses used in our studies framed within the scope of clinically relevant doses humans is an important point. While the recommended dosage of pitavastatin for hypercholesterolemia is 1-4 mg daily, healthy humans tolerate up to 64 mg¹², which is comparable to 1 mg/kg and the dose that we used in our animal studies.

3. Background information regarding statins and melanoma are notably missing from the manuscript. This is not the first report showing melanoma sensitivity to statins. In addition, statin-use has been shown to be associated with better outcome in melanoma, a concept that should be incorporated into the manuscript.

We apologize for not introducing sufficient background on prior work relating to the potential use of statins in the treatment of melanoma within the original submission. Relevant description and references are highlighted in yellow.

4. Legends and text are incomplete, which forces the reader to turn to materials and methods to better understand the experiment being performed. Provide clear description of all experiments in the text and/or legend so reader can easily follow along. E.g. Fig 1e; e.g. providing dose and time of pitavastatin used throughout all experiments, along with rationale for these choices.

We have in this revised manuscript provided better experimental descriptions within the text, legends, and Experimental procedures. As an example, we have detailed that "Pitavastatin has been used previously at 1 or 3 mg/kg¹³, at 4 or 8 mg/kg¹⁴. We used pitavastatin at 1 mg/kg b.i.d.; and for the purpose of effective tumor growth inhibition, future experiments will need to investigate the pharmacokinetics and higher pitavastatin dosages.

5. Several examples of figure labels that are difficult to read, missing or misaligned with the data, including Fig 1f, Ext Fig 1c, Ext Fig 6d, can't see protein labels within Fig 4a volcano plot (right); Ext Fig 10b, Ext Fig 8b, and Ext Fig 10a.

The labels have been corrected and made more legible in the revised figures.

6. Figures are not in order in the text, e.g. change Ext Fig 3d, 3e to 3b, 3c to have figures in chronological order in text; similarly Ext Fig 4d, and Ext Fig 6c.

The figures are now presented and labelled according to the flow of the text, as it should have been.

7. Can't read labels highlighted in blue, Ext Fig 4d. This figure should be Ext Fig 4a as the mevalonate pathway is introduced in the text prior to the data with the mevalonate metabolites.

We have changed to gray color, and it is now found as **Extended data Fig. 4a**.

8. This statement "Since GGPP is mainly used to prenylated protein substrates..." is not supported by references that provide data showing that the majority of GGPP is used for protein isoprenylation compared to other GGPP end-products such as CoQ and dolichol. I don't know of such reference(s). Thus, this statement may be unfounded and need to be rewritten.

We have rewritten and corrected this section/statement and now include adequate references.

9. Authors 'need to check this': see text "Tumor xenograft studies using melanoma cell lines (NEED TO CHECK THIS!!!!)"

We apologize that the reminder remained; it has been deleted after making sure the text is correct.

10. Text states A375P and K029A, yet data pertaining only to the latter is shown. Show A375P data too. "In agreement, chronic treated BRAF-inhibitor resistant A375P and A melanoma cells exhibited reduced expression of these two RAB genes (Fig. 3e)."

This is shown in **Extended Fig. 5a**; and now detailed as such.

11. As MITF is investigated in the results section, the role of MITF in melanoma and its relationship with PGC1alpha needs to be mentioned in the introduction as Nature Communications has a broad readership that may not be familiar with MITF-PGC1alpha. Similarly, an intro on the RAB family of proteins would also help support readers, which is particularly important as so much of the work focuses on the RABS. Same could be said for integrins, FAK, etc.

We have introduced new text to address this point (page 152-153; 189-190).

12. Statistical analysis of Fig 3j is needed. Are these differences statistically significant?

When we use GraphPad to analyze drug inhibition data, only IC50 and 95% CI (profile likelihood) are provided. Such as for the knockdown of RAB6B, the data is (GI50 (shScramble) 4.2µM, (range 3.68 to 4.9 µM) compared to GI50 (shRAB6B) 1.30µM, (range 1.1 to 1.55)).

13. Ext Fig 6b: what are the relative growth kinetics of K029A-R-RAB6B/27A WT vs MT? Is pitavastatin just targeting the faster growing tumor?

K029A-R-RAB6B/27A WT and MT grow at similar rates both in vitro and in vivo. A375P tumors (**Extended Fig. 3o**) grow faster than A375P-R tumors (**Fig. 2h**), however, pitavastatin showed its effect on A375P-R tumors instead of A375P tumors, suggesting pitavastatin is not just targeting the faster growing tumor.

14. Ext Fig 7b: which cells are being evaluated here?

Now Ext Fig 6b. K029A chronic treated BRAF-inhibitor adapted cells was used and that is now clearly in the legend.

Reference:

- 1 Luo, C. *et al.* H3K27me3-mediated PGC1 α gene silencing promotes melanoma invasion through WNT5A and YAP. *The Journal of clinical investigation* **130**, 853-862 (2020).
- 2 Luo, C. *et al.* A PGC1 α -mediated transcriptional axis suppresses melanoma metastasis. *Nature* **537**, 422-426, doi:10.1038/nature19347 (2016).
- 3 Jiang, P. *et al.* In vitro and in vivo anticancer effects of mevalonate pathway modulation on human cancer cells. *Br J Cancer* **111**, 1562-1571, doi:10.1038/bjc.2014.431 (2014).
- 4 Thai, L. *et al.* Farnesol is utilized for isoprenoid biosynthesis in plant cells via farnesyl pyrophosphate formed by successive monophosphorylation reactions. *Proceedings of the National Academy of Sciences of the United States of America* **96**, 13080-13085, doi:10.1073/pnas.96.23.13080 (1999).
- 5 Rink, J. S. *et al.* Targeted reduction of cholesterol uptake in cholesterol-addicted lymphoma cells blocks turnover of oxidized lipids to cause ferroptosis. *J Biol Chem* **296**, 100100, doi:10.1074/jbc.RA120.014888 (2021).
- 6 Bergamini, C., Moruzzi, N., Sblendido, A., Lenaz, G. & Fato, R. A water soluble CoQ10 formulation improves intracellular distribution and promotes mitochondrial respiration in cultured cells. *PLoS One* **7**, e33712, doi:10.1371/journal.pone.0033712 (2012).
- 7 Schumacher, M. M. & DeBose-Boyd, R. A. Posttranslational Regulation of HMG CoA Reductase, the Rate-Limiting Enzyme in Synthesis of Cholesterol. *Annu Rev Biochem* **90**, 659-679, doi:10.1146/annurev-biochem-081820-101010 (2021).
- 8 Schafer, Z. T. *et al.* Antioxidant and oncogene rescue of metabolic defects caused by loss of matrix attachment. *Nature* **461**, 109-113, doi:10.1038/nature08268 (2009).
- 9 Piskounova, E. *et al.* Oxidative stress inhibits distant metastasis by human melanoma cells. *Nature* **527**, 186-191, doi:10.1038/nature15726 (2015).
- 10 Deng, Z., Wang, H., Liu, J., Deng, Y. & Zhang, N. Comprehensive understanding of anchorage-independent survival and its implication in cancer metastasis. *Cell Death Dis* **12**, 629, doi:10.1038/s41419-021-03890-7 (2021).
- 11 Muller, P. A. *et al.* Mutant p53 drives invasion by promoting integrin recycling. *Cell* **139**, 1327-1341, doi:10.1016/j.cell.2009.11.026 (2009).
- 12 Catapano, A. L. Pitavastatin: a different pharmacological profile. *Clin Lipidol* **7**, 3-9 (2012).
- 13 Hu, W. & Jiang, W. B. Pitavastatin-attenuated cardiac dysfunction in mice with dilated cardiomyopathy via regulation of myocardial calcium handling proteins. *Acta Pharm* **64**, 105-115, doi:10.2478/acph-2014-0004 (2014).
- 14 Wang, L. *et al.* Pitavastatin slows tumor progression and alters urine-derived volatile organic compounds through the mevalonate pathway. *FASEB J* **33**, 13710-13721, doi:10.1096/fj.201901388R (2019).

REVIEWERS' COMMENTS

Reviewer #1 (Remarks to the Author):

Authors have addressed all my concerns and the other reviewers' too.

Reviewer #3 (Remarks to the Author):

In this revised version of their manuscript, the authors have incorporated and experimentally addressed the main issues raised before by this reviewer.

Regarding our major point 1, whether BRAFi-resistant cell lines with silenced PPARGC1A are indeed more aggressive than BRAFi-resistant cells displaying increased PPARGC1A expression, the authors chose to address this point by in vitro experiments (migration assays), rather than in vivo, as requested. Obviously, an in vivo approach would have been more suitable to support the conclusion provided in lines 82-84, so I suggest to slightly tone down the statement made there (e.g. by stating "the data suggest..." or "are in line with the hypothesis...").

Regarding our major point 2 (the main findings appear to apply only to a subset of melanoma cells), the authors have now included a third cell line in their analyses to make their point. That's fine, but the corresponding paragraph is a bit confusing (lines 65ff, starting with 'five lines were assessed...'; reference to Extended Data Fig. 1a is unclear in this context). The authors should try to clarify in the paragraph that suppression of PPARGC1A in emerging resistant cells is not the common, unique scenario but only happens in a subset of cells.

Reviewer #4 (Remarks to the Author):

In general, the authors have been responsive and have added new data to address my previous concerns. There are additional minor issues, however the manuscript is now sound and sufficiently coherent.

REVIEWERS' COMMENTS

Reviewer #1 (Remarks to the Author):

Authors have addressed all my concerns and the other reviewers' too.

- *We are delighted to learn that our now revised manuscript was found responsive to the salient and well-thought critiques.*

Reviewer #3 (Remarks to the Author):

In this revised version of their manuscript, the authors have incorporated and experimentally addressed the main issues raised before by this reviewer.

Regarding our major point 1, whether BRAFi-resistant cell lines with silenced PPARGC1A are indeed more aggressive than BRAFi-resistant cells displaying increased PPARGC1A expression, the authors chose to address this point by in vitro experiments (migration assays), rather than in vivo, as requested. Obviously, an in vivo approach would have been more suitable to support the conclusion provided in lines 82-84, so I suggest to slightly tone down the statement made there (e.g. by stating “the data suggest...” or “are in line with the hypothesis...”).

- *Experiments were included for both in vitro (migration) and in vivo (metastasis) across the three cell lines A375P, K029A, and SKMEL3 that display reduced PPARGC1A expression following chronic treatment with BRAF inhibitor. In addition, we have now changed the text (lines 87-89) following the reviewer’s suggestion to more appropriately read: Collectively, these results suggest that in a subset of melanoma cells, chronic BRAF inhibitor treatment can cause epigenetic adaptation that involves PPARGC1A suppression and acquisition of aggressive melanoma traits.*

Regarding our major point 2 (the main findings appear to apply only to a subset of melanoma cells), the authors have now included a third cell line in their analyses to make their point. That’s fine, but the corresponding paragraph is a bit confusing (lines 65ff, starting with ‘five lines were assessed...’; reference to Extended Data Fig. 1a is unclear in this context). The authors should try to clarify in the paragraph that suppression of PPARGC1A in emerging resistant cells is not the common, unique scenario but only happens in a subset of cells.

- *We have revised the text accordingly and now made it clear that in two out of the five melanoma cell lines PPARGC1A expression increased during chronic BRAF-inhibitor treatment. Lines (67-69) now read: Chronic PLX4032 treated G361 and SKMEL5, however, yielded resistant cells with elevated PPARGC1A expression (Extended Data Fig. 1c), which parallels the other end of the change-in-expression continuum seen across melanoma biopsies pre-/post-BRAF inhibitor treatment (Fig 1a).*

Reviewer #4 (Remarks to the Author):

In general, the authors have been responsive and have added new data to address my previous concerns. There are additional minor issues, however the manuscript is now sound and sufficiently coherent.

- *We are grateful for the prior critique which prompted changes and inclusions that certainly improved our manuscript.*